# Gadd45B Deficiency Drives Radio-Resistance in BRAF^V600E^-Mutated Differentiated Thyroid Cancer by Disrupting Iodine Metabolic Genes

**DOI:** 10.3390/cancers17193201

**Published:** 2025-09-30

**Authors:** Shan Jiang, Zhiwen Hong, Qianjiang Wu, Rouhan A, Zhaobo Wang, Xue Guan, Xinghua Wang, Ari A. Kassardjian, Yali Cui, Tengchuang Ma

**Affiliations:** 1Department of Nuclear Medicine, Harbin Medical University Cancer Hospital, Harbin 150081, China; 202201470@hrbmu.edu.cn (S.J.); 202401563@hrbmu.edu.cn (Z.H.); arh2000@163.com (R.A.); wzb990208@163.com (Z.W.);; 2Institute of Cancer Prevention and Treatment, Harbin Medical University Cancer Hospital, Harbin 150081, China; 2022021633@hrbmu.edu.cn; 3Animal Laboratory Center, The Second Affiliated Hospital of Harbin Medical University, Harbin 150081, China; guanxue2019@163.com; 4Department of Radiation Oncology, City of Hope National Medical Center, 1500 E. Duarte Rd, Duarte, CA 91010, USA; akassardjian@coh.org

**Keywords:** radioactive iodine-refractory differentiated thyroid cancer, radioactive iodine therapy, Gadd45B, MAP3K4, MYCBP

## Abstract

Radioactive iodine (RAI) is a cornerstone therapy for differentiated thyroid cancer, but many tumours—especially those with the BRAF^V600E^ mutation—become RAI-refractory and stop taking up iodine. We analysed patient tissues and public datasets and found that Gadd45B is consistently reduced in RAI-refractory disease. Using thyroid cancer cell lines and mouse models (including patient-derived xenografts), we show that restoring Gadd45B re-sensitises tumours to RAI, increases uptake, and slows growth. Mechanistically, Gadd45B modulates two complementary axes: it interacts with MAP3K4 to dampen MAPK signalling, and it restrains MYCBP–c-Myc–TERT activity. Together, these effects upregulate iodine-handling genes (e.g., NIS, TPO, Tg) and improve tumour differentiation. Clinically, low Gadd45B correlates with poor outcomes, supporting its potential as a biomarker and therapeutic target. While intratumoural recombinant Gadd45B showed benefit in vivo, we did not directly confirm its cellular entry; future studies will test delivery strategies and validate safety in patients.

## 1. Introduction

DTC accounts for over 90% of thyroid malignancies and continues to rise in incidence globally. According to GLOBOCAN 2020 data, thyroid cancer ranks as the ninth-most common malignancy worldwide, with approximately 586,000 new cases annually, and the highest incidence in Eastern Asia and parts of Southern Europe. DTC includes papillary thyroid carcinoma (PTC), follicular thyroid carcinoma (FTC), and Hürthle cell carcinoma (HCC), with PTC being the most prevalent subtype (~80–85%). Standard treatment for DTC includes total thyroidectomy followed by RAI therapy and thyroid-stimulating hormone suppression [1,2].

The role of postoperative radioactive iodine continues to expand due to the increasing incidence of DTC. However, transformation to RAIR-DTC reduces the therapeutic efficacy of RAI treatment and represents a significant cause of mortality and treatment failure [3,4], which is characterized by the loss of iodine avidity in metastatic lesions, progressive disease despite RAI uptake, or the presence of lesions that do not concentrate RAI on diagnostic or post-therapy scans. Clinically, RAIR-DTC is defined by any of the following: (1) absence of RAI uptake in all lesions, (2) disease progression within 12–16 months after RAI therapy, or (3) mixed uptake with progression in non-RAI–avid sites. RAIR-DTC accounts for approximately 10–15% of all DTC cases and is associated with poor prognosis, with a 10-year survival rate of <10% once distant metastases occur [5,6]. Emerging treatment strategies aim to address the challenges posed by RAIR-DTC [7,8]. Phase I/II clinical trials using MLN0128 (NCT02244463) and interventional trials using AZD6244 (NCT00970359) selectively target and restore iodine metabolism in the tumor microenvironment of RAIR-DTC. Modifications to the tumor microenvironment promote RAI reuptake by RAIR-DTC [9,10]. Moreover, BRAF^V600E^ and MEK inhibitors, such as dabrafenib and trametinib, enhance RAI uptake in RAIR-DTC patients by targeting the MAPK signaling pathway [11,12]. Despite their efficacy, these treatment strategies ultimately fail to resolve the fundamental challenges associated with RAIR-DTC, partly due to low structural and biochemical remission rates and unsatisfactory RAI uptake rates [13,14]. Consequently, further research is essential to elucidate the mechanisms underlying RAIR-DTC development and to identify precise targets and comprehensive treatment strategies.

Recent studies have implicated stress response genes, including the Gadd45 family, in modulating cancer cell survival, DNA repair, and metabolic adaptation—features potentially relevant to RAI resistance [15]. Growth arrest and DNA damage protein 45 beta (Gadd45B) plays a crucial role in cellular stress responses by regulating the cell cycle, DNA repair, and apoptosis [16]. Previous studies have shown that it may exert an inhibitory effect on cancer development and function as a tumor suppressor [17,18]. Specifically, low Gadd45B expression correlates with a more aggressive tumor phenotype and increased metastatic propensity in triple-negative breast cancer [19]. By facilitating DNA repair and inhibiting cell cycle progression, Gadd45B suppresses clonogenic proliferation in breast cancer cells [20,21]. Moreover, the Gadd45 family can enhance breast cancer cell sensitivity to chemotherapeutic drugs, further inhibiting tumor growth [22]. In non-small-cell lung cancer (NSCLC), Gadd45B activates the p38 MAPK signaling pathway, inducing cell cycle arrest and inhibiting NSCLC cell proliferation [23,24]. Gadd45B also sensitizes NSCLC tumor cells to radiotherapy, thereby improving treatment efficacy [25]. In colorectal cancer, Gadd45B expression closely correlates with tumor staging and patient survival rates [26]. It inhibits colorectal cancer cell proliferation by promoting apoptosis and suppressing angiogenesis [27]. Simultaneously, Gadd45B modulates the Wnt/β-catenin signaling pathway [28,29]. The function of Gadd45B may vary across different cancer subtypes due to its interactions with specific cellular microenvironments, tumor genotypes, and signaling pathways [30]. However, the role of Gadd45B in thyroid cancer physiology and its relationship with RAIR-DTC remains poorly understood.

In this study, we explored Gadd45B deficiency as a potential driver of RAI treatment resistance and characterized its mechanism of action in cell cycle progression. We collected clinical samples of BRAF^V600E^-mutated RAIR-DTC and DTC for RNA sequencing. All samples exhibited BRAF^V600E^ mutation. Notably, Gadd45B expression in RAIR-DTC tissue was significantly lower than in DTC, potentially exacerbating abnormal MAPK signaling pathway activation and promoting RAIR-DTC development. At the molecular level, overexpression of Gadd45B reduced MAPK pathway activity [31]. This reduction increased the expression of key iodine metabolic genes, including NIS, TPO, and Tg.

Furthermore, Gadd45B competed with MYCBP for c-Myc binding and mediated MYCBP ubiquitination and degradation. This disrupted the stability of the c-Myc–MYCBP complex, thereby reducing c-Myc oncogenic activity. Consequently, this process influenced TERT gene promoter activity and further enhanced iodine metabolism gene expression, aiding in reversing the dedifferentiated state of RAIR-DTC and restoring its ability to absorb radioactive iodine. A treatment strategy focused on Gadd45B could hold substantial clinical value and provide a scientific basis for translation into clinical practice.

## 2. Materials and Methods

### 2.1. Study Design

The objective of this study was to investigate the mechanism of dedifferentiation in radioactive iodine-refractory differentiated thyroid cancer (RAIR-DTC). We screened Gadd45B, a non-critical gene with the potential to affect key pathways, using sequencing technology (n = 16, next-generation sequencing, NSG). Its oncogenic effects were confirmed through analyses of various universal databases. Subsequently, phenotyping experiments were conducted to assess the oncogene-suppressive effects of Gadd45B in both BRAF^V600E^-mutated and wild-type thyroid cancer cells.

Furthermore, we evaluated the potential of Gadd45B to inhibit thyroid cancer dedifferentiation through its interacting factors, validated via in vivo experiments. All experiments in this study were repeated at least three times to demonstrate biological reproducibility and ensure sufficient statistical power for comparisons. The study was unblinded, and statistical methods were not employed to predetermine sample size. Details of the in vivo experiments, including the number of cells used, duration, and statistical tests, are described in the figure legends. No formal a priori sample size calculation was performed; however, sample sizes were determined based on previous studies in the field that yielded statistically robust results. Post hoc power analysis was conducted for the primary endpoints, confirming that the study was sufficiently powered (power > 0.8) to detect the observed effect sizes. Blinding of investigators was not feasible for all experiments due to the nature of the interventions; nevertheless, bias was minimized by random allocation of samples/animals to treatment groups, independent data acquisition by separate personnel, and blinded analysis of outcome measures where possible.

### 2.2. Human Thyroid Tissue

Two independent cohorts of patients with differentiated thyroid cancer (DTC) who underwent thyroidectomy at the Harbin Medical University Cancer Hospital Otorhinolaryngology Department were included in this study. In this study, RAIR-DTC was defined according to the criteria recommended by the 2015 American Thyroid Association (ATA) Guidelines [32] and recent consensus statementsCohort 1 consisted of 16 patients treated for differentiated thyroid cancer (DTC) at our institution between January 2018 and January 2023. The samples were categorized into three groups:

Normal thyroid tissue (N, n = 5): histologically normal thyroid tissue collected from patients diagnosed with DTC in whom the primary tumor was very small, allowing for the safe sampling of adjacent non-tumorous thyroid tissue during surgery.

DTC group (n = 6): primary tumor tissue obtained at initial thyroidectomy from patients diagnosed with DTC who had not received any prior therapy.

RAIR-DTC group (n = 5): tumor tissue obtained from patients initially diagnosed with DTC, who underwent total thyroidectomy followed by ^131^I treatment, and subsequently developed progressive disease characterized by either loss of radioiodine uptake or enlargement of metastatic lesions. These RAIR-DTC samples were obtained via surgical resection or core needle biopsy of recurrent or metastatic lesions.

All specimens were immediately snap-frozen in liquid nitrogen or fixed in 4% paraformaldehyde. Each sample was independently reviewed and graded by two board-certified pathologists. Transcriptome sequencing of these tissues was performed by Novogene Co., Ltd. (Beijing, China) to identify differentially expressed genes (DEGs) among the N, DTC, and RAIR-DTC groups.Cohort 2 included tumor tissues and para-tumor tissues from 5 patients with DTC who underwent resection between January 2018 and January 2023.Cohort 3 consisted of 176 pathological diagnosed with thyroid cancer who underwent surgical treatment in the Department of Head and Neck Surgery at Harbin Medical University Cancer Hospital between May 2022 and October 2023. Clinical data included age, sex, tumor size, histological subtype, TNM stage, and radioactive iodine (RAI) treatment history. Written informed consent was obtained from all participants, and the study was approved by the institutional ethics committee.Cohort 4 comprised 121 BRAFV600E-mutated DTC patients, selected from the 176 patients originally included in Cohort 3.Patient-derived xenografts were established from freshly resected BRAF^V600E^-mutated RAIR-DTC tumor specimens obtained from patients who had undergone thyroidectomy and failed RAI therapy. Tumor donors included 5 females and 2 males, aged 38–65 years, with histological subtypes encompassing 5 papillary and 2 poorly differentiated carcinomas. All tumors exhibited reduced or absent NIS expression by IHC, and at least one iodine metabolism gene mutation (e.g., SLC5A5, TPO). The overall engraftment rate was 71% (5/7), with a median latency period of ~4 weeks before palpable tumors formed. Histopathological examination confirmed that the PDXs retained the original tumor’s morphological features, BRAF^V600E^ mutation status, and heterogeneous expression of NIS, Tg, and TPO, indicating that these models faithfully recapitulate the molecular and phenotypic heterogeneity of the donor tumors.

### 2.3. Ethical Statement

This study was approved by the Ethics Committee of the Harbin Medical University Cancer Hospital (KY2024-94), and all tissue samples were obtained with informed consent.

### 2.4. Cell Line and Drug Treatment

Human thyroid cancer cell lines BCPAP, K1, and TPC-1 were obtained from Miaoling Biological Co., Ltd. (Dalian, China). All cell lines were routinely cultured at 37 °C in RPMI 1640 or DMEM medium supplemented with 10% fetal bovine serum. For specific experiments, cells were treated with 1 μm BRAF kinase inhibitor PLX4720 (Selleck Chemicals, Houston, TX, USA), 1 μm MEK1/2 inhibitor AZD6244 (Selleck Chemicals, TX, USA), 100 mg/mL cycloheximide (HY-12320, MCE), Gadd45B recombinant protein (HY-P70212, MCE), or 10 nm MG-132 (HY-13259, MCE) for the specified durations. All inhibitors were dissolved in dimethyl sulfoxide (DMSO), aliquoted, and stored at −80 °C until use. Equivalent DMSO volumes were used as controls.

According to the International Cell Line Authentication Committee (ICLAC) database, the K1 cell line is contaminated with the GLAG-66 cell line. While this contamination did not alter the BRAF^V600E^ mutation status or the MAPK-dependency of the cells, we cannot exclude the possibility that transcriptional or functional heterogeneity introduced by GLAG-66 cells may have influenced some downstream analyses. All functional assays were repeated in an independently validated BRAF^V600E^-mutated DTC cell line (BCPAP) and results were concordant, supporting the robustness of our main conclusions. Nevertheless, readers should interpret the K1-derived data with this caveat in mind. Both lines originate from human papillary thyroid carcinoma (PTC). Our genetic analysis confirmed that the BCPAP and K1 cell lines used in this study exhibited typical heterozygous BRAF^V600E^ and TERT C228T mutations. Cells were authenticated via short tandem repeat (STR) analysis and tested for mycoplasma contamination. The K1 cell line met the purpose of this study to investigate the roles of BRAF^V600E^ and TERT promoter mutations in human cancer.

RNA interference, lentivirus transfection, and expression plasmids were utilized. Control siRNAs and specific siRNAs targeting Gadd45B (human), MAP3K4 (human), MYCBP (human), and c-Myc (human) were obtained from Gene Pharma (Shanghai, China). Cells at ~50% confluence were transfected with 50 nM siRNAs using Lipofectamine 2000 (Invitrogen, NY, USA). Lentiviruses expressing shRNA targeting MAP3K4 and control lentiviruses were purchased from HanBio Biotechnology Co., Ltd. (Shanghai, China). Cells were infected at a multiplicity of infection (MOI) ranging from 30% to 60% confluence, followed by selection with 2 μg/mL puromycin for one week and maintenance screening at 1 μg/mL for an additional week. Infection efficiency was validated using qRT-PCR was performed using an ABI 7500 Real-Time PCR System (Applied Biosystems, Foster City, CA, USA).and Western blotting. Recombinant human Gadd45B protein was applied at the indicated concentrations for 48 h in vitro or administered intraperitoneally every other day for 2 weeks in animal experiments. The purity of the recombinant Gadd45B protein (>95%) was confirmed by SDS–PAGE and Coomassie blue staining. Biological activity was validated by assessing its ability to modulate MAP3K4 phosphorylation in TPC-1 and BCPAP cells. Endotoxin levels were measured using the Limulus amebocyte lysate (LAL) assay and confirmed to be <0.1 EU/μg.

Gadd45B gene knockout and knockdown was performed using a CRISPR/Cas9-mediated genome editing approach, following previously described protocols. Single-guide RNAs (sgRNAs) targeting human Gadd45B were designed and synthesized by Miaoling. The targeting sequences were described as follows: shGadd45B #H1: GACCTGTCTTTGCGAAAGCAA; sgRNA1: 5′-GGAGGCTGGGACGCTGCGGA-3′ The sgRNA was cloned into the PX459 vector (Addgene, Watertown, MA, USA) and transfected into thyroid cancer cells using Lipofectamine 3000 (Thermo Fisher Scientific, Waltham, MA, USA) according to the manufacturer’s instructions. Overexpression pCMV-GADD45B-3×FLAG-Neo NM_015675.4 Forty-eight hours after transfection, cells were selected with puromycin (2 μg/mL) for 1–2 weeks to establish stable knockout lines. Successful gene disruption was confirmed by Sanger sequencing and Western blot analysis.

### 2.5. Mice

Four-week-old female BALB/c nude mice and four-week-old NOD SCID female mice were purchased from Beijing Vital River Laboratory Animal Technologies. The mice were group-housed in specific pathogen-free (SPF) conditions with a 12 h light/12 h dark cycle (150–300 lux), an ambient temperature of 20–26 °C, and a humidity range of 40–70%, with air ventilation four times per hour. Mice were randomly assigned to experimental groups based on body weight and were fed a standard chow diet. All animal experiments were approved by the Institutional Animal Care and Use Committee (IACUC) of Harbin Medical University Cancer Hospital. For all animal experiments, investigators responsible for measuring tumor volume and assessing RAI uptake were blinded to the group allocation throughout the study period. Randomization codes were held by a separate technician not involved in data collection or analysis, and unblinding was performed only after completion of statistical analyses. Humane endpoints were set at 2000 mm^3^ tumor volume or 20% weight loss; buprenorphine (0.05 mg kg^−1^ s.c.) was given post-surgery, and animals were monitored daily with tumor size measured twice weekly.

To avoid physiological uptake by the native thyroid gland during the ^131^I uptake experiments in mice, the following procedure was applied. Mice received gastric perfusion of potassium iodide solution (200 μL/day) for three consecutive days prior to SPECT imaging, as previously described.

### 2.6. Data Collection and Processing

Gene expression data, clinical information, and single nucleotide variation data for thyroid carcinoma were obtained from multiple sources, including the Genomic Data Commons (https://portal.gdc.cancer.gov/), GEO database (https://www.ncbi.nlm.nih.gov/), cBioPortal (https://cbioportal.org/), and Timer2.0 (http://timer.cistrome.org/) all accessed on 1 January 2000. Initial normalization and log2 conversion of the original data were performed using the Million Transcripts per Million (TPM) method. TCGA sample inclusion criteria were limited to type 01A (Primary Tumor) samples with complete survival information.

### 2.7. Silver Stain-Guided In-Gel Digestion and LC–MS/MS Identification

Protein complexes were immunoprecipitated using anti-Flag agarose from TPC-1 and Bcpap cells stably overexpressing Flag-Gadd45B. Eluates were resolved by SDS–PAGE and visualized by silver staining (Pierce Silver Stain Kit) according to the manufacturer’s protocol. Bands of interest were excised with a sterile scalpel, destained (fresh 30 mM K_3_ [Fe(CN)_6_]/100 mM Na_2_S_2_O_3_, 1:1, 5–10 min), reduced with 10 mM DTT (56 °C, 45 min) and alkylated with 55 mM iodoacetamide (RT, dark, 30 min). In-gel digestion was performed overnight at 37 °C with sequencing-grade trypsin (Promega, Madison, WI, USA).

Peptides were extracted, dried, and reconstituted for nano-LC–MS/MS (EASY-nLC coupled to an Orbitrap-type mass spectrometer). Data-dependent acquisition was used (typical settings: MS1 resolution 60,000–120,000; HCD MS2; dynamic exclusion enabled). RAW files were searched in Proteome Discoverer (v2.x)/MaxQuant (v1.x) against the UniProt human reference proteome with the following parameters: enzyme trypsin, max 2 missed cleavages; carbamidomethyl-C fixed; oxidation-M and acetyl-protein N-term variable; precursor tolerance ±10 ppm; fragment tolerance 0.02 Da. Peptide-spectrum matches and protein IDs were filtered to FDR ≤ 1% (target–decoy). Proteins were reported only if supported by ≥2 unique peptides. Under these criteria, peptides mapped uniquely to GADD45B and MYCBP in the corresponding gel bands.

### 2.8. RNA Extraction and Quantitative Real-Time PCR (qRT-PCR)

Total RNA was extracted from cultured cells using the TRIzol reagent (#15596-018; Ambion, Life Technologies, Carlsbad, CA, USA) and reverse-transcribed to cDNA using the SuperScript III First-Strand Synthesis System (#18080-051; Invitrogen, Life Technologies, Carlsbad, CA, USA). Gene expression analysis was performed in triplicate using FastStart Universal SYBR Green Master with ROX (#04913850001; Roche Applied Science, Indianapolis, IN, USA) on an Applied Biosystems 7900HT Fast Real-Time PCR System. Relative gene expression was calculated using the 2^−ΔΔCt^ method, with GAPDH used as the internal control for normalization. Primers of Gadd45B: (forward) 5′-TGTACGAGTCGGCCAAGTTG-3′ and 5′-ATTTGCAGGGCGATGTCATC-3′ (reverse). The TERT promoter region was amplified by PCR using the primers 5′-AGTGGATTCGCGGGCACAGA-3′ (forward) and 5′-CAGCGCTGCCTGAAACTC-3′ (reverse). The BRAF^V600E^ mutation hotspot was amplified using the primers 5′- GGCAGAGTGCCTCAAAAAGAA-3′ (forward) and 5′- GCGGTGAATTTTTGGCAATG-3′ (reverse). The PCR products were subjected to Sanger sequencing to detect BRAF^V600E^ and TERT promoter mutations.

### 2.9. Western Blotting

Proteins were extracted from thyroid cancer tissues and cells using a protein extraction kit (Beyotime, Shanghai, China). Extracted proteins were separated by 10% SDS-PAGE (Beyotime) and transferred to PVDF membranes (Roche, Shanghai, China). Membranes were blocked with blocking buffer (Beyotime) for 20 min, incubated overnight at 4 °C with primary antibodies diluted in buffer, and then incubated with HRP-conjugated secondary antibodies (Agilent Technologies) for 1 h at room temperature. Protein bands were visualized using an ECL kit (YEASEN, Shanghai, China) and a chemiluminescent gel imaging system (Vilber, Paris, France). Antibodies used are presented in Appendix A. Densitometric analysis of Western blot bands was performed using ImageJ software (version 1.53 k, National Institutes of Health, Bethesda, MD, USA). Band intensities were normalized to the corresponding GAPDH (or other indicated loading control) signals. Background subtraction was applied using the “rolling ball” method in ImageJ to eliminate non-specific background. All values were expressed as relative intensity (arbitrary units) compared with control samples.

### 2.10. Cell Proliferation Assays

For the Cell Counting Kit-8 (CCK-8) assays, transfected cells were seeded in 96-well plates (500 cells/well). At designated time points (1, 2, 3, 4, 5, and 6 days), 10 μL of CCK-8 solution was added to each well, followed by incubation at 37 °C for 2 h. Absorbance at 450 nm was measured using a Multiskan GO plate reader (Thermo Fisher Scientific).

For 5-ethynyl-2′-deoxyuridine (EdU) assays, transfected cells were seeded in 96-well plates (20,000 cells/well) and cultured for 2 days. Subsequently, 50 μM EdU (RiboBio, Guangzhou, China) was added to each well, and cells were incubated at 37 °C for 2 h. Cells were then fixed, permeabilized, and stained with 400 μL Hoechst 33342 to visualize nuclei. The proportion of EdU-positive cells was determined by counting cells in five randomly selected areas per well.

For colony formation assays, transfected cells were seeded in 6-well plates (500 cells/well) and incubated in 2 mL of culture medium for 10 days. Colonies were fixed with alcohol, stained with crystal violet, and quantified.

### 2.11. Flow Cytometry

Tumor cells were cultured in serum-free basal medium for flow cytometry analysis. To detect cell apoptosis, cells were rinsed, resuspended in ice-cold phosphate-buffered saline (PBS), and incubated with Annexin V-FITC and propidium iodide (PI) staining solution (apoptosis detection kit; Vazyme) at 37 °C for 20 min.

### 2.12. Transwell Assays

For Transwell migration assays, 200 μL of serum-free medium containing 2 × 10^4^ transfected cells were seeded into the upper chamber of a Transwell plate (Corning, NY, USA), while 800 μL of serum-containing medium was added to the lower chamber. After 24 h of incubation, cells in the upper chamber were fixed, stained with crystal violet (Beyotime) for 20 min, and observed using an inverted optical microscope (Olympus, Tokyo, Japan). Cell counts were determined from three randomly selected fields of view. To evaluate cell invasion ability, Matrigel was applied to the Transwell plate surface prior to seeding.

### 2.13. Immunofluorescence Assays

Microarrays of thyroid cancer tissues and sections from mouse xenografts were prepared for immunohistochemical staining. Slides were incubated in 3% H_2_O_2_ for 5 min at room temperature to block endogenous peroxidase activity, followed by antigen retrieval in sodium citrate buffer for 15 min at 95 °C. After blocking with 5% normal goat serum for 10 min, slides were incubated overnight at 4 °C with primary antibodies. The following day, slides were incubated with appropriate secondary antibodies for 1 h at room temperature. Nuclei were visualized with DAPI (Beyotime, China) staining, and images were captured using a fluorescence microscope (Olympus, Japan).

For immunofluorescence, cells attached to slides were fixed with 4% paraformaldehyde and permeabilized using Immunostaining Permeabilization Buffer with Saponin (Beyotime, China). Slides were blocked with 5% bovine serum albumin (BSA) in PBS for 1 h at room temperature and incubated overnight at 4 °C with the appropriate primary antibody. Following three PBS washes, slides were incubated with corresponding secondary antibodies for 1 h at room temperature. Nuclei were stained with DAPI, and images were obtained using a fluorescence microscope.

### 2.14. RAI Uptake Assays

For in vitro assays, cells were seeded into 6-well plates, incubated with 1 μCi ^125^I for 2 h, washed twice with pre-cooled PBS, digested with trypsin, and collected for analysis. Cell radioactivity was measured using a gamma radioimmunoassay counter (GC-1500, ZONKIA, China).

For in vivo assays, mice were fed water containing 0.1% potassium iodide (Sangon, A100512) for 7 days before the ^125^I uptake experiments. Each mouse received a tail vein injection of 1 mCi ^125^I, and subcutaneous tumors were assessed for counts per minute using a gamma counter.

### 2.15. Dual-Luciferase Reporter Assay

The C228T/C250T-mutant and wild-type TERT-pGL4.10 luciferase reporter plasmids were purchased from Abxin (China). Luciferase activity was measured following standard protocols. Briefly, cells were seeded at 70% confluence and transfected with pRL-TK and either C228T-TERT-Luc or C250T-TERT-Luc plasmids using X-treme GENE HP DNA transfection reagent (Roche) in 12-well plates.

For siRNA experiments, cells were transfected with specific siRNAs 24 h before luciferase plasmid transfection. For trametinib treatment, cells were treated with 1 μM PLX4720 and AZD6244 or DMSO for 24 h after luciferase plasmid transfection. In overexpression experiments, cells were co-transfected with Sp1-WT or Sp1-T739A expression plasmids and luciferase plasmids.

Cells were collected 48 h post-treatment, and luminescence intensity was measured using an EnSpire Multimode Plate Reader (PerkinElmer) with the dual-luciferase reporter assay system (Promega), following the manufacturer’s instructions. Data were normalized to pRL-TK luciferase activity. All experiments were performed in triplicate.

### 2.16. Co-IP

Co-IP experiments were performed to investigate the interaction between Gadd45B and MAP3K4. TPC-1 and Bcpap thyroid cancer cells stably overexpressing Gadd45B were generated using lentiviral transduction of the pCMV-GADD45B-3×FLAG-Neo construct, followed by neomycin (G418) selection. For Co-IP, these stable cell lines were further transfected with pcDNA3.1-MAP3K4-His plasmid using Lipofectamine 3000 (Thermo Fisher Scientific) to achieve transient overexpression of His-tagged MAP3K4.

After 48 h, cells were lysed in IP lysis buffer (Beyotime, China) supplemented with protease and phosphatase inhibitors. Lysates were incubated overnight at 4 °C with either anti-Flag antibody (Sigma-Aldrich, St. Louis, MO, USA) or anti-His antibody (Abcam, Cambridge, UK), followed by 2 h incubation with protein A/G agarose beads (Thermo Fisher Scientific). Immunoprecipitates were washed three times with lysis buffer and eluted by boiling in SDS loading buffer. Western blotting was then performed using reciprocal antibodies (anti-His for Flag-IP, anti-Flag for His-IP) to confirm the protein–protein interaction.

### 2.17. Molecular Docking Analysis

Molecular docking analysis was conducted using rigid protein–protein docking to investigate the interaction between Gadd45B and MAP3K4 using GRAMM-X (http://gramm.compbio.ku.edu/). Structural domains of Gadd45B and MAP3K4 were sourced from the AlphaFold Protein Structure Database (https://alphafold.ebi.ac.uk/). Protein–protein interactions were visualized and analyzed using PyMOL (Version 3.0) and PDBePISA (https://www.ebi.ac.uk/pdbe/pisa/) all accessed on 1 January 2000. Docking parameters were set according to the software default scoring function, with the grid box centered on the predicted active site. The exhaustiveness parameter was set to 8, and all ligands were subjected to energy minimization before docking. The docking procedure followed best practice guidelines as described in previous research with binding poses ranked by predicted binding affinity (kcal/mol) and visually inspected for interaction plausibility.

### 2.18. IP-MS

Immunoprecipitation coupled with mass spectrometry (IP-MS) was performed to analyze proteins extracted from transfected cells. Immunoprecipitation was carried out using primary antibodies and Protein A/G agarose beads (Santa Cruz Biotechnology, Shanghai, China). The isolated immunoprecipitates were subjected to mass spectrometry analysis using a Thermo Scientific mass spectrometer (Waltham, MA, USA). Peptide labeling was conducted using Tandem Mass Tag (TMT) 10-plex reagents according to the manufacturer’s instructions. Mass accuracy was set to ±10 ppm for precursor ions and ±0.02 Da for fragment ions. Peptide identification required at least one unique peptide per protein, and peptide-spectrum matches were filtered at a false discovery rate (FDR) of 1% using the target-decoy strategy. Database searching was performed against the UniProt human reference proteome using *Proteome Discoverer* software (version 2.5) with carbamidomethylation (C) as a fixed modification and oxidation (M) and TMT labeling as variable modifications.

### 2.19. Immunohistochemistry

Tissue sections were deparaffinized in xylene, dehydrated through a graded ethanol series, and subjected to antigen retrieval in sodium citrate solution (Servicebio, Wuhan, China) using a heater. The slides were incubated with the primary antibody overnight at 4 °C, followed by incubation with the secondary antibody at 25 °C for 2 h. Signals were developed using 3,3′-diaminobenzidine (DAB) solution (Servicebio).

The intensity and staining area were scored as follows:Staining intensity scores: 0 (negative), 1 (weak), 2 (moderate), 3 (strong).Staining area scores: 0 (0–10%), 1 (11–25%), 2 (26–50%), 3 (51–75%), 4 (76–100%).

The two scores were multiplied to calculate the IHC staining score.

### 2.20. Statistical Analysis

Data presentation and statistical rationale. All in vitro results are reported as mean ± standard deviation (SD) to reflect technical variability among replicates, whereas in vivo and ex vivo data are presented as mean ± standard error of the mean (SEM) to estimate population-level precision. Normality was verified with the Shapiro–Wilk test; equal variances were checked with Levene’s test. For two-group comparisons we used two-tailed Student’s *t*-test; for three or more groups we applied one- or two-way ANOVA followed by Holm-Šídák’s multiple-comparison test because it preserves family-wise error rates without excessive loss of power. Kaplan–Meier survival curves were compared by the Mantel–Cox log-rank test, with Bonferroni correction applied when pairwise survival analyses exceeded two groups. Statistical significance is denoted by *p* * < 0.05, *p* ** < 0.01, *p* *** < 0.001 and *p* **** < 0.0001.

## 3. Results

### 3.1. Gadd45B as a Key Regulator of BRAF^V600E^-Mutated Thyroid Cancer Dedifferentiation

Based on analysis of the TCGA-THCA dataset, the MAPK signaling pathway exhibited a significant correlation with gene mutations, notably the BRAFV600E mutation, which is the most prevalent mutation in thyroid cancer [33]. Among various mutation types, single-nucleotide polymorphisms (SNPs) were the most frequent, with C-to-A transitions being the predominant single nucleotide variants (SNVs). Consistent with previous reports [34], the BRAFV600E mutation accounted for a substantial proportion of thyroid cancer cases, with V600E being the major hotspot. (Appendix A). To identify key genes involved in thyroid cancer progression, we collected BRAF^V600E^-mutated thyroid cancer tissues and corresponding normal tissues from patients who underwent ^131^I treatment at Harbin Medical University Cancer Hospital, categorizing the tissues into DTC and RAIR-DTC groups (Figure 1A). RNA profiling identified differentially expressed genes (DEGs) from three groups of samples (cohort 1) (Figure 1B). Among the DEGs, Gadd45B emerged as a significantly downregulated gene during RAIR-DTC formation (Figure 1C). Further functional analysis of DEGs using Gene Ontology (GO) and Kyoto Encyclopedia of Genes and Genomes (KEGG) databases revealed that the MAPK signaling pathway was significantly enriched in thyroid cancer with downregulated Gadd45B expression alongside BRAF^V600E^ mutation (Figure 1D). The key biological processes, cellular components, and molecular functions affected by Gadd45B downregulation included protein catabolic processes, intracellular protein complexes, and protein-macromolecule adaptor activity (Figure 1E). As DTC progressed to RAIR-DTC, Gadd45B expression levels decreased, with similar trends confirmed in external datasets GSE104005 [35] and GSE53157 [36] (Figure 1F and Appendix A). Additionally, an inverse correlation was observed between Gadd45B expression levels and BRAF^V600E^ mutations, indicating a potential association between Gadd45B dysregulation and RAIR-DTC formation (Figure 1G).

We hypothesized that Gadd45B contributes to the development of multiple malignant tumor phenotypes and promotes thyroid cancer dedifferentiation. Pan-cancer analysis revealed that Gadd45B expression was downregulated across various cancer types, with a particularly notable reduction in thyroid cancer. These findings supported the hypothesis that decreased Gadd45B expression facilitates tumor progression and plays a pro-tumorigenic role (Figure 1H and Appendix A). Moreover, low Gadd45B expression in thyroid cancer with BRAF^V600E^ mutations was confirmed through the TCGA-THCA and Timer2.0 databases (Figure 1I,J). Analysis also demonstrated that Gadd45B expression correlated with different pathological subtypes of papillary thyroid carcinoma (Appendix A). Ten pairs of cancerous and corresponding para-cancer tissues (cohort 2) were collected to validate Gadd45B protein expression. Immunohistochemistry (IHC) and Western blot (WB) results confirmed its downregulation in BRAF^V600E^-mutated thyroid cancer (Figure 1K,L). Correlation analysis revealed that Gadd45B expression was significantly associated with iodine uptake-related genes, including TSHR, Tg, THRA, and TPO (Appendix A). A network model further illustrated the BRAF^V600E^–Gadd45B axis and its associated pathways in thyroid cancer (Appendix A). Subsequent correlation analysis of Gadd45B IHC scores from cohort 3 indicated that its expression was negatively associated with BRAF^V600E^ mutations (Figure 1M). Patients with low Gadd45B expression exhibited significantly shorter progression-free intervals (PFI) and disease-free intervals (DFI), indicating a worse prognosis (Figure 1N,O). These results confirmed that Gadd45B is significantly downregulated and functions as a pivotal regulator in the dedifferentiation of BRAF^V600E^-mutated thyroid cancer, contributing to tumor progression and poor prognosis.

### 3.2. Gadd45B Deficiency Reduce RAI Uptake and Increase the Malignancy in BRAF^V600E^-Mutated Thyroid Cancer

To further verify Gadd45B deficiency in BRAF^V600E^-mutated thyroid cancer, we assessed its endogenous expression in the Bcpap and K1 cell lines, which harbor the BRAF^V600E^ mutation, and in the BRAF wild-type thyroid cancer cell line TPC-1. TPC-1, a well-characterized differentiated thyroid carcinoma cell line with RET/PTC rearrangement and wild-type BRAF, was selected as the BRAF-wildtype comparator. This cell line exhibits relatively preserved NIS expression and differentiation markers, making it an appropriate reference model for assessing the effects of BRAF^V600E^ mutation and Gadd45B regulation on iodine metabolism. Protein levels of Gadd45B were significantly lower in Bcpap (0.17-fold) and K1 (0.19-fold) compared to TPC-1 (Figure 2A). Subsequently, we generated Gadd45B knockdown, knockout, and overexpression models in Bcpap, K1, and TPC-1 cell lines to evaluate their function (Appendix A–C). Gadd45B overexpression suppressed the cell proliferation rate, while its knockdown and knockout increased the proliferative capacity of Bcpap (KO: from 0.24% ± 0.03% to 0.39% ± 0.0.6%, OE:0.18% ± 0.04% to 0.15% ± 0.03%) and K1 (KO: from 0.22% ± 0.04% to 0.41% ± 0.1%, OE: 0.21% ± 0.05% to 0.18% ± 0.03%. However, this effect was not observed in TPC-1 (KO: from 0.18% ± 0.03% to 0.19% ± 0.0.6%, OE: 0.21% ± 0.05% to 0.18% ± 0.03%) (Figure 2B,C and Appendix A). These findings were further confirmed through colony formation assays (Figure 2D). Migration assays revealed that Gadd45B deficiency enhanced the migration capabilities of Bcpap (KO: from 294 ± 28 to 485 ± 53, OE:367 ± 43 to 298 ± 34) and K1 (KO: from 311 ± 58 to 505 ± 38, OE: 322 ± 21 to 229 ± 39), with no significant changes in TPC-1 (there was no statistically significant difference in changes between the groups) (Figure 2E,F and Appendix A). Additionally, Gadd45B deficiency significantly inhibited the apoptosis rate, decreasing it from 13.1% ± 0.9% to 8.4% ± 0.9%, whereas Gadd45B sufficiency robustly induced apoptosis, increasing it from 14.5% ± 0.8% to 24.3% ± 1.6% in Bcpap and K1. In contrast, Gadd45B expression showed minimal effects on apoptosis in TPC-1 (Figure 2G,H). The expression and functional levels of NIS directly affect the efficiency of RAI uptake [37,38]. Therefore, we investigated the effects of Gadd45B knockout and overexpression on NIS expression. Immunofluorescence assays showed that Gadd45B knockout decreased NIS levels, while Gadd45B overexpression increased NIS expression (Figure 2I). We used NaClO_4_ as a competitive inhibitor to validate NIS-mediated iodide uptake. NaClO_4_ is a well-established inhibitor that competes with iodide for NIS transport. To minimize concerns about off-target effects, we used vehicle-treated controls and matched experimental conditions to ensure observed changes were specifically attributable to NIS-mediated uptake inhibition. As shown in Figure 2J, the presence of Gadd45B enhanced RAI uptake in Bcpap and K1 cells, but not in TPC-1 cells. In Bcpap cells, RAI uptake decreased from 7916 ± 998 to 3990 ± 1232 cpm/10^6^ cells upon Gadd45B knockdown, whereas overexpression increased uptake to 12,550 ± 2303 cpm/10^6^ cells. Similarly, in K1 cells, uptake decreased from 7916 ± 932 to 4490 ± 902 cpm/10^6^ cells with Gadd45B knockdown, and increased to 18,550 ± 2803 cpm/10^6^ cells with overexpression. The addition of NaClO_4_ reversed the differences in ^125^I uptake among these groups, confirming that Gadd45B influences RAI uptake by modulating NIS function. To assess the role of Gadd45B in tumorigenesis, we conducted in vivo experiments using xenograft tumor models to evaluate its effect on the dedifferentiation process in BRAF^V600E^-mutated thyroid cancer (Figure 2K). Gadd45B overexpression significantly suppressed the growth rate and tumor volume in Bcpap (increase from 211 ± 23 mm^3^ to 291 ± 17 mm^3^ when knockoff Gadd45B, increase to 155 ± 24 mm^3^) xenografts TPC-1 xenografts (Figure 2L,M,N and Appendix A). Subsequent gamma counting revealed that Gadd45B overexpression increased tumor radioactivity in Bcpap xenografts, further supporting the in vitro observations (Figure 2O,P). Based on these results, we preliminarily concluded that Gadd45B is a key factor that enhances RAI uptake and regulates the dedifferentiation process in BRAF^V600E^-mutated thyroid cancer.

### 3.3. Increased Gadd45B Expression in BRAF^V600E^-Mutated Thyroid Cancer Reverses the Process of Dedifferentiation via the MAPK Signaling Pathway

Based on preliminary results, we hypothesized that Gadd45B functions as a bypass inhibitor of the MAPK signaling pathway (Figure 3A). Gadd45B overexpression negatively regulated MEK and ERK phosphorylation, which are terminal components of the MAPK signaling pathway (Figure 3B). As shown in Figure 3C, treatment with 0.5 μM PLX4720 inhibited BRAF^V600E^ expression over time and increased Gadd45B expression. Selumetinib (AZD6244), a U.S. FDA-approved MEK inhibitor, reduced p-MEK levels in a time-dependent manner at 1 μM concentration but did not affect Gadd45B expression (Figure 3D). BRAF^V600E^ acted as an upstream repressor of Gadd45B, while PLX4720 upregulated Gadd45B expression (Figure 3E). Gadd45B knockout offset the p-MEK downregulation induced by AZD6244 (Figure 3F). These results suggested that Gadd45B functions as an intermediary between BRAF^V600E^ and downstream components of the MAPK signaling pathway. The combination of Gadd45B overexpression with PLX4720 or AZD6244 significantly reduced p-MEK (3.6-fold vs. control, *p* < 0.01), and p-ERK levels (3.6-fold vs. control, *p* < 0.01) (Figure 3G,H). Furthermore, combined treatment with Gadd45B overexpression, PLX4720, and AZD6244 markedly inhibited MAPK signaling and increased the expression of NIS, Tg, and TPO (Figure 3I,J), further suppression of p-MEK and p-ERK (reduction by ~65% compared to control), as compared to each monotherapy (*p* < 0.05 for all comparisons). Immunofluorescence assays showed that the combined treatment group displayed reduced p-MEK and p-ERK fluorescence intensity in Bcpap cells (Figure 3K) and K1 cell lines (Appendix A). Figure 3L demonstrates that combined treatment significantly enhanced NIS expression, particularly in the cellular membrane region. After combined treatment, BRAF^V600E^ expression was downregulated, while Gadd45B expression was upregulated (Appendix A).

To investigate the effects of combined treatment on tumor growth and RAI uptake, xenograft tumor models were established (Figure 3M). Surveillance results revealed that combined treatment effectively inhibited tumor growth (Figure 3N,O). Western blotting and histological immunostaining further validated that combined treatment increased the expression of NIS, Tg, and TPO (Figure 3P,Q). Consistently, RAI uptake was upregulated in the combined treatment group at specific time points (Figure 3R and Appendix A). These findings, consistent with gamma counting results, demonstrated that simultaneous Gadd45B overexpression combined with PLX4720 and AZD6244 increased iodine metabolic gene expression, providing a mechanism for enhanced iodine uptake.

### 3.4. Synergistic Effects of MAP3K4 and Gadd45B Overexpression Suppress Tumor Growth and Enhance Radioiodine Uptake in Xenograft Models

MAP3K4 plays a pivotal role in cellular processes such as apoptosis and autophagy. Additionally, MAP3K4 interacts with key partners, including Gadd45, to influence cell growth and development. Its enzyme activity is inhibited in the absence of Gadd45 binding [39,40,41]. As shown in Figure 4A, molecular docking predicted a binding affinity of −7.6 kcal/mol between Gadd45B and MAP3K4. Key residues critical for the interaction were identified, offering structural insights into the binding mechanism. Specifically, hydrogen bonds were formed between residues TYR 141 of Gadd45B and GLU 143 of MAP3K4, as well as between ARG 91 and THR 153. Co-immunoprecipitation (Co-IP) and silver staining experiments further revealed a stable interaction between Gadd45B and MAP3K4 (Figure 4B,C). Overexpression of Gadd45B upregulated MAP3K4 expression in Bcpap cells (Figure 4D). Immunofluorescence (IF) analysis showed that MAP3K4 accumulated more abundantly in Gadd45B-overexpressing Bcpap cells compared to the control group (Figure 4E). To validate these results, we generated MAP3K4 knockdown and overexpression Bcpap cell lines (Appendix A). Knockdown of MAP3K4 counteracted the inhibitory effects of Gadd45B on the MAPK signaling pathway, whereas MAP3K4 overexpression restored these effects (Figure 4F,G). Notably, overexpression of both MAP3K4 and Gadd45B demonstrated synergistic inhibition of the MAPK signaling pathway and significantly stimulated the expression of NIS, Tg, and TPO (Figure 4H,I). Immunofluorescence results corroborated these findings (Figure 4J,K). To evaluate the effects of Gadd45B and MAP3K4 overexpression in vivo, two independent xenograft model cohorts were established (Figure 4L). One cohort was used for tumor volume surveillance, while the other was used to measure RAI uptake. Combined treatment resulted in a significant reduction in tumor volume and an increase in RAI uptake (Figure 4M,N). Additionally, MAP3K4 plays a pivotal role in apoptosis signal transduction. Protein expression levels of NIS, Tg, TPO, and MAP3K4-related apoptotic proteins, including Caspase-8, P38, and P53, were also upregulated (Figure 4O,P). IHC results demonstrated consistent findings (Appendix A). In summary, the synergistic effects of MAP3K4 and Gadd45B suppressed MAPK signaling pathway activity, which subsequently upregulated proteins related to iodine metabolism and apoptosis, leading to reduced tumor growth and enhanced RAI uptake.

### 3.5. Deficiency of Gadd45B Stabilizes MYCBP-c-MYC Complex and TERT Promoter Activity by Alleviating MYCBP Degradation Stress

Recent studies have indicated that BRAF^V600E^ mutation is widely associated with TERT promoter mutations in human cancers, particularly in thyroid cancer and melanoma [42,43,44]. Immunoprecipitation-mass spectrometry (IP-MS) analysis of Gadd45B-overexpressing Bcpap and K1 cells revealed potential Gadd45B-interacting proteins, among which MYC-binding protein (MYCBP) showed an inverse correlation with Gadd45B (Figure 5A,B and Appendix A). MYCBP interacts with c-MYC, enhancing its ability to activate E-box-dependent transcription. MYCBP is typically located in the cytoplasm but translocates to the nucleus during the S phase of the cell cycle, where it binds c-MYC [45,46].

We speculated that Gadd45B affects the stability of the MYCBP–c-MYC complex, stimulating TERT promoter activity and promoting thyroid cancer dedifferentiation. Co-IP and silver staining assays confirmed the interaction between Gadd45B and MYCBP (Figure 5C,D). Gadd45B colocalized with MYCBP, and its overexpression downregulated MYCBP protein levels (Figure 5E and Appendix A). We hypothesized that Gadd45B regulates MYCBP stability in a proteasome-mediated manner. Gadd45B overexpression significantly impaired MYCBP stability in the presence of the protein synthesis inhibitor cycloheximide (CHX) (Figure 5F). Furthermore, the proteasome inhibitor MG132 reversed Gadd45B-induced MYCBP degradation, confirming that Gadd45B regulates MYCBP stability through proteasomal degradation (Figure 5G). Consistent with these results, MYCBP ubiquitination was enhanced by Gadd45B upregulation (Figure 5H). Overexpression of MYCBP reversed the Gadd45B-induced downregulation of c-MYC and TERT protein expression, indicating that MYCBP serves as a critical connector between Gadd45B and TERT (Figure 5I). Furthermore, the c-MYC-specific inhibitor miR-26b-5p reversed the increased c-MYC and TERT expression induced by Gadd45B knockout (Figure 5J,K and Appendix A). Additionally, MYCBP-specific inhibition by MYCi975 and MYCi361 decreased TERT protein levels, with MYCi975 reversing TERT overexpression induced by MYCBP overexpression (Appendix A–I). Studies have shown that increased TERT promoter activity not only upregulates TERT gene expression but is also significantly associated with poor prognosis. To investigate this further, luciferase reporter assays were conducted to assess TERT promoter activity. Both the wild-type and the C228T- or C250T-mutated TERT promoters responded to Gadd45B overexpression as well as miR-26b-5p or c-MYC inhibition. Notably, TERT promoters harboring C228T or C250T mutations were more sensitive compared to the wild-type (Figure 5L and Appendix A). Moreover, the upstream transcription factors of the TERT gene showed significant reduction (Figure 5M,N). Combined Gadd45B overexpression and MYCBP downregulation significantly suppressed the MAPK signaling pathway and TERT expression while upregulating iodine metabolism-associated genes (Figure 5O–R). Our study demonstrated that Gadd45B deficiency stabilizes the MYCBP–c-MYC complex and enhances TERT promoter activity by alleviating MYCBP degradation stress. These findings underscore the pivotal role of Gadd45B in regulating the stability of the MYCBP–c-MYC complex and TERT promoter activity.

### 3.6. Synergistic Therapeutic Efficacy of Gadd45B Overexpression with MAP3K4 and MYCBP Inhibition Direct to Thyroid Cancer Treatment

To further validate the synergistic therapeutic efficacy of MAP3K4 and MYCBP inhibition in conjunction with Gadd45B overexpression, we established an iodine uptake rate model in mice (Figure 6A). The combined treatment of Gadd45B and MAP3K4 overexpression, along with MYCBP inhibition, significantly inhibited tumor proliferation and prolonged mouse survival rates (Figure 6B,C). Various administration methods did not impose any additional survival burden on the mice (Figure 6D). Radioactive iodine uptake in the combined treatment group increased significantly at 24, 48, and 72 h (Figure 6E), indicating that the combined treatment positively restored iodine metabolic function. IHC assays confirmed that the combined treatment significantly increased the percentage of NIS-, Tg-, and TPO-positive cells while decreasing MYCBP- and c-MYC-positive cells in tumor tissues derived from xenografts (Figure 6F and Appendix A). Western blotting results were consistent with these findings (Figure 6G and Appendix A). BIBR1532, a selective telomerase inhibitor, can regulate thyroid cancer dedifferentiation in conjunction with the BRAF^V600E^ inhibitor PLX4720. We hypothesized that Gadd45B overexpression enhances the therapeutic efficacy of these inhibitors. To test this, we constructed an iodine uptake rate model in mice treated with PLX4720 and BIBR1532 (Figure 6H). Combined treatment with Gadd45B overexpression, PLX4720, and BIBR1532 significantly inhibited tumor proliferation and prolonged mouse survival rates (Figure 6I,J). These treatments did not impose any additional survival burden on the mice (Figure 6K). The restored iodine uptake rate further indicated that the combined treatment positively influenced iodine metabolism (Figure 6L). IHC assays confirmed a significant increase in NIS-, Tg-, and TPO-positive cells and a decrease in BRAF^V600E^- and c-MYC-positive cells in xenograft tumor tissues (Figure 6M and Appendix A). Corresponding Western blotting assays provided similar results (Figure 6N and Appendix A). The two mouse iodine uptake rate models validated the synergistic therapeutic effects of Gadd45B overexpression combined with mechanical and clinical target-based treatments. Combined treatment significantly reduced tumor proliferation, improved survival rates, and enhanced iodine metabolism without imposing additional survival burdens. These findings suggest that Gadd45B overexpression enhances the therapeutic efficacy of these inhibitors, providing a theoretical basis for its application in RAIR-DTC treatment with RAI.

### 3.7. Collaborative Efficacy of Gadd45B Protein with BRAF^V600E^ Inhibition and TERT Promoter Inhibition Reverse the Dedifferentiation Process of RAIR-DTC

We developed patient-derived xenograft (PDX) models using RAIR-DTC tissues to further assess the impact of combining PLX4720 and BIBR1532 with recombinant Gadd45B protein on iodine uptake restoration and dedifferentiation reversal in RAIR-DTC (Figure 7A). Two distinct PDX model assays were employed to validate the efficacy of recombinant Gadd45B protein. The recombinant Gadd45B protein significantly inhibited tumor growth and increased survival rates in the PDX models, while depletion of Gadd45B counteracted these effects (Figure 7B–D). Importantly, recombinant Gadd45B protein did not impose any additional burden on other organs (Appendix A). Given the critical role of iodine uptake in RAI treatment, we assessed radioactivity in a separate PDX model group administered with ^125^I, following the previously described workflow. Consistently, recombinant Gadd45B protein enhanced radioiodine accumulation in tumors, whereas the anti-Gadd45B antibody reduced this effect (Figure 7E). Additionally, recombinant Gadd45B protein significantly improved the therapeutic efficacy of PLX4720 and BIBR1532, which inhibited tumor growth when administered simultaneously. Analyses of body weight and survival curves yielded consistent results (Figure 7F–H). The ^125^I uptake rate in thyroid tumors further increased with the combined treatment (Figure 7I). Administration of recombinant Gadd45B protein markedly enhanced the inhibitory effects of PLX4720 on BRAF^V600E^ and BIBR1532 on the TERT promoter (Figure 7J,K). Protein expression levels of NIS, Tg, and TPO were upregulated by the combined treatment, emphasizing its impact on reversing RAIR-DTC dedifferentiation (Figure 7L,M). IHC images corroborated the findings presented in Figure 7J,N. Previous studies have established a correlation between Gadd45B loss and unfavorable prognoses in thyroid cancer patients. To further investigate the clinical implications of our findings, we performed immunohistochemical staining to examine the interactions between Gadd45B and its associated proteins, as well as their relationship to dedifferentiation, using specimens from a cohort of 121 BRAF^V600E^-mutated thyroid cancer patients (cohort 4). Univariate and multivariate logistic regression analyses showed that low Gadd45B expression was an independent predictor of RAIR-DTC (Figure 7O and Appendix A). Compared to DTC, RAIR-DTC exhibited lower expression levels of MAP3K4 and higher expression levels of MYCBP (Figure 7P). Our analysis revealed a positive correlation between MAP3K4 and P53 expression with Gadd45B, while MYCBP and c-MYC showed a negative correlation (Figure 7Q). Statistical evaluation of IHC scores confirmed the significance of these associations (Figure 7R). For the diagnostic evaluation of Gadd45B, receiver operating characteristic (ROC) curve analysis was performed using both our clinical cohort and TCGA-THCA dataset. An optimal cutoff value of 0.42 (normalized expression units) was determined by Youden’s index, yielding a sensitivity of 81.6% and a specificity of 78.4% for distinguishing RAIR-DTC from RAI-sensitive DTC. The area under the ROC curve (AUC) was 0.856 (95% CI: 0.785–0.914), indicating good discriminatory power. Similar performance metrics were observed in the independent validation cohort (AUC = 0.842, sensitivity = 79.3%, specificity = 76.1%). These findings position Gadd45B as a biomarker for predicting potential RAI treatment benefits in thyroid cancer patients, highlighting its role as both a diagnostic indicator and a therapeutic target for RAIR-DTC.

## 4. Discussion

RAI therapy has a long-standing history of successful application in the treatment of thyroid cancer. However, dedifferentiated thyroid cancer that does not respond to RAI therapy tends to have a poor prognosis. Modifying gene expression represents a promising strategy for converting RAI-refractory thyroid cancer into RAI-sensitive thyroid cancer. Currently, most studies have focused on individual signaling proteins rather than examining the broader spectrum of signaling pathways involved [47]. As indicated by previous clinical trials (NCT04462471), single-agent treatment may be insufficient to achieve durable redifferentiation efficacy due to the reactivation of signaling pathways and the subsequent development of drug resistance. This underscores the potential benefits of combination therapies involving targeted agents and RAI. Additionally, a comprehensive investigation into the molecular mechanisms of thyroid cancer can facilitate the identification of novel targets and the development of innovative therapies, providing safer, more effective, and cost-efficient treatment options for RAIR-DTC patients.

Our transcriptome sequencing of para-cancer tissue, DTC, and RAIR-DTC revealed a strong negative correlation between the BRAF^V600E^ mutation and Gadd45B expression (Figure 1A–F). Reduced Gadd45B expression contributed to the progression from DTC to RAIR-DTC in the context of the BRAF^V600E^ mutation (Figure 2J–P and Figure 3M,N). The role of Gadd45B loss in promoting dedifferentiation suggests that a comprehensive investigation into the interplay between BRAF^V600E^ mutation and Gadd45B expression, as well as its underlying mechanisms, could enhance the development of targeted treatment strategies for RAIR-DTC.

Initial results from a clinical trial indicated that GSK2118436 (Dabrafenib, a BRAF^V600E^ inhibitor) improved the stable disease (SD) rate, achieving a 70% response based on RECIST criteria. However, thyroglobulin levels increased in 60% of participants (NCT04462471). In another clinical trial, the administration of AZD6244 resulted in a 60% increase in RAI uptake among patients. Nevertheless, this treatment did not achieve complete remission, with a partial response (PR) rate of only 62.5% (NCT00970359). Independent use of MAPK signaling pathway inhibitors may not fully restore RAI uptake in RAIR-DTC patients, as abnormal activation of alternative targets within the MAPK pathway can result in unfavorable outcomes. MAP3K4 is a critical molecule in this pathway. Interaction coefficients from docking analysis revealed a strong interaction between Gadd45B and MAP3K4 (Figure 4A–D). This interaction was further validated through experiments demonstrating that Gadd45B and MAP3K4 collaborate to activate the p38-p53 pathway, leading to increased expression of iodine metabolic proteins. Furthermore, overexpression of Gadd45B in conjunction with MAP3K4 overexpression effectively restored iodine uptake (Figure 4N). Recent studies have highlighted the involvement of the TERT promoter in the dedifferentiation process of thyroid cancer [48]. To further investigate Gadd45B targets, we analyzed its interactions using IP-MS (Figure 5B). Our study confirmed that Gadd45B overexpression enhanced MYCBP ubiquitination while suppressing c-MYC expression (Figure 5H). It has been proposed that c-MYC exerts pro-oxidant functions in thyroid cancer by increasing the probability of TERT promoter mutations. Based on this theoretical framework, we investigated the effects of Gadd45B overexpression on c-MYC expression and TERT promoter activity. Our results showed that Gadd45B reduced MYCBP expression by increasing MYCBP ubiquitination, which diminished c-MYC’s impact on promoting TERT promoter mutations and thereby impeded thyroid cancer dedifferentiation (Figure 5L,Q). Subsequently, a tumor-burden mouse model validated the synergistic effects of Gadd45B overexpression with MAP3K4 overexpression and MYCBP inhibition (Figure 6A–G), as well as with classical targeted drugs PLX4720 and BIBR1532 (Figure 6H–N). Furthermore, PDX models of RAIR-DTC showed consistent results that supported our findings (Figure 7H,I,L,M). Correlations between related markers and dedifferentiation status were further substantiated by evidence from cohort 4 (Figure 7O–Q). These results suggest that Gadd45B could serve as a significant biomarker for predicting patients likely to benefit from RAI treatment and as a potential therapeutic target for synergistic treatment in RAIR-DTC. However, this study has several limitations. This study aimed to elucidate the mechanisms underlying Gadd45B function in RAIR-DTC and to determine whether regulating Gadd45B expression could reverse the dedifferentiation process. Notably, our study did not address Gadd45B’s role in cell cycle regulation during tumor initiation or mechanisms of tumor progression, including tumor cell autophagy, epithelial–mesenchymal transition (EMT), immune escape, and drug resistance development. Future research is necessary to explore Gadd45B’s role in the immunotherapy microenvironment, potential synergistic functions of other Gadd45 family members, and strategies to safely and effectively enhance Gadd45B expression during immunotherapy. The expression of Gadd45B remains a fundamentally and clinically relevant question that warrants further investigation in RAIR-DTC. These modifications may impact tumor development and provide a potential adjuvant treatment strategy to synergize with RAI therapy. Based on our integrated functional, mechanistic, and preclinical data, the MAP3K4–MYCBP–c-MYC axis emerges as the most promising candidate for next-stage validation. This axis not only represents a downstream effector pathway of Gadd45B deficiency but also directly links MAPK signaling perturbations to impaired iodine metabolism and enhanced radio-resistance. Given that MAP3K4 restoration partially rescues NIS expression and sensitizes BRAFV600E-mutated RAIR-DTC cells to RAI therapy in vitro and in vivo, targeting this pathway—potentially in combination with recombinant Gadd45B protein or BRAF/TERT inhibition—may offer a rational translational strategy. Future work will focus on validating this axis in larger patient cohorts and assessing therapeutic efficacy in immunocompetent models. Although intratumoral administration of recombinant Gadd45B protein significantly enhanced RAI uptake and suppressed tumor growth in vivo, the precise mechanism by which the recombinant protein exerts its effects remains to be elucidated. As Gadd45B is not a secreted protein, the possibility of cellular internalization and its intracellular activity require further validation. In the present study, we did not perform direct assays, such as Western blotting or immunofluorescence, to confirm cellular uptake of the recombinant protein. Therefore, while our data support a therapeutic effect of recombinant Gadd45B, future studies should rigorously assess whether and how the exogenous protein penetrates tumor cells and mimics endogenous Gadd45B function.

In future studies, priority should be given to validating Gadd45B-based adjuvant strategies in preclinical immunotherapy models, given their strong mechanistic rationale and the urgent clinical need for effective radiosensitizers in RAIR-DTC. This will be followed by pilot clinical studies with safety and pharmacokinetic endpoints, enabling early translational assessment. Parallel efforts will explore the MAP3K4–MYCBP–c-MYC axis as a complementary target, with the goal of integrating molecular restoration strategies into multimodal treatment regimens. Such a phased approach will allow systematic evaluation from mechanistic insight to clinical applicability.

## 5. Conclusions

In summary, our study demonstrates that Gadd45B deficiency is a key driver of radioiodine resistance in BRAFV600E-mutated RAIR-DTC, primarily through disruption of iodine metabolism and sustained activation of the MAP3K4–MYCBP–c-MYC axis. Importantly, restoration of Gadd45B expression, either alone or in combination with BRAF or MEK inhibitors, significantly re-sensitized resistant tumors to RAI therapy in preclinical models, including patient-derived xenografts. These findings not only elucidate a novel mechanistic basis for RAIR-DTC progression but also nominate Gadd45B as a compelling therapeutic target with direct clinical applicability. Our work suggests that Gadd45B-based strategies—such as targeted gene therapy or pharmacological reactivation—could potentially restore RAI avidity in refractory DTC, thereby offering a new avenue for precision medicine. Future efforts will focus on validating these approaches in advanced translational models and ultimately initiating early-phase clinical trials to assess the efficacy of Gadd45B modulation in RAIR-DTC patients.

## Figures and Tables

**Figure 1 cancers-17-03201-f001:**
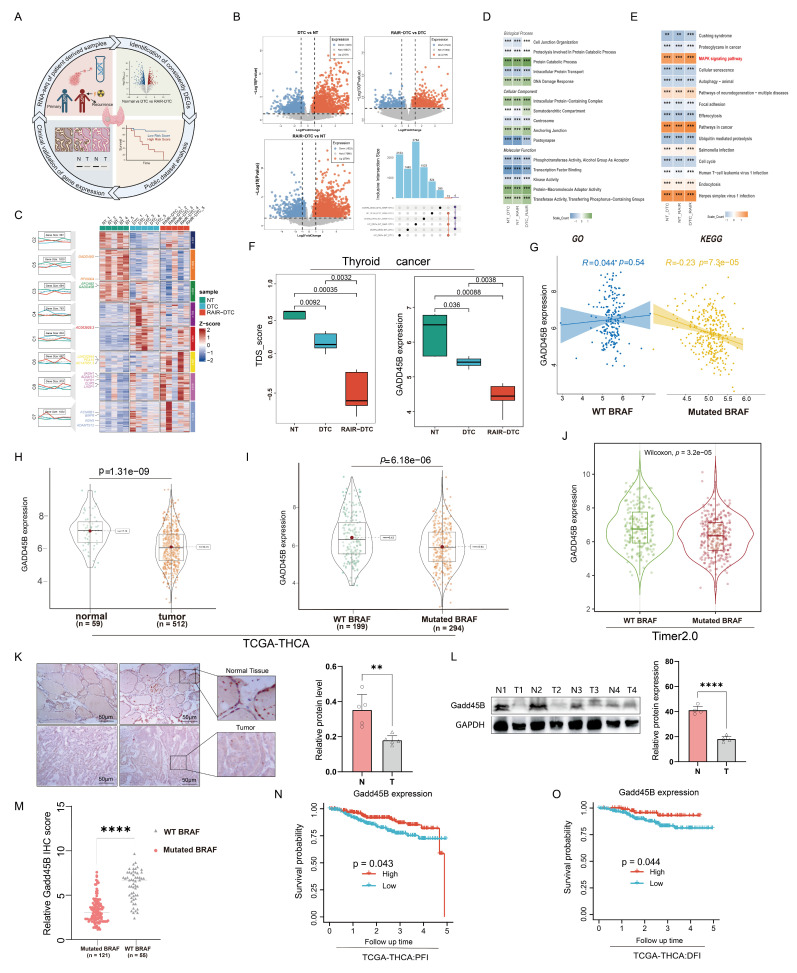
Gadd45B deficiency drive the formation of RAIR-DTC. (**A**) Workflow for the identification and preliminary validation of candidate genes responsible for RAIR−DTC development. (**B**) Differentially expressed genes (DEGs) among normal tissue, DTC, and RAIR−DTC based on RNA−seq data from cohort 1. (**C**) Significantly upregulated or downregulated DEGs and correlations of gene modules among normal tissue, DTC, and RAIR−DTC. (**D**) Gene Ontology (GO) analysis showing the biological process, cellular component, and molecular function enrichment of DEGs. (**E**) KEGG pathway analysis revealing cancer-related signaling pathway enrichment. (**F**) Variability in Gadd45B expression levels and Thyroid Differentiation Score (TDS) with disease progression from normal tissue to DTC and RAIR−DTC. (**G**) Correlation of Gadd45B expression levels between BRAF wild-type and BRAFV600E-mutated thyroid cancer. (**H**) Gadd45B expression levels in normal tissue versus thyroid cancer (data sourced from TCGA−THCA; *p* = 1.31 × 10^−9^). (**I**) Comparison of Gadd45B expression levels in BRAF wild-type versus BRAFV600E-mutated thyroid cancer (data derived from TCGA−THCA; *p* = 6.18 × 10^−6^). (**J**) Gadd45B expression levels in BRAF wild-type and BRAF^V600E^−mutated thyroid cancer (data collected from Timer2.0; Wilcoxon test, *p* = 3.2 × 10^−5^). (**K**) Representative images of Gadd45B IHC staining in normal tissue and thyroid cancer tissue, with quantitative analysis from cohort 2 (n = 5). Magnification: 40 × 10; scale bars: 50 μm. (**L**) Protein expression levels of Gadd45B in normal tissue and thyroid cancer tissue, with relative quantitative analysis (n = 5). (**M**) Relative Gadd45B IHC scores in BRAF wild−type and BRAFV600E-mutated thyroid cancer (cohort 3, n = 186). (**N**,**O**) Progression-free interval (PFI) and disease−free interval (DFI) curves based on Gadd45B-related survival probabilities (data collected from TCGA−THCA). Data are shown as mean ± SD. Statistical significance: ** *p* < 0.01, *** *p* < 0.001, **** *p* < 0.0001. The uncropped blots are shown in Appendix A.

**Figure 2 cancers-17-03201-f002:**
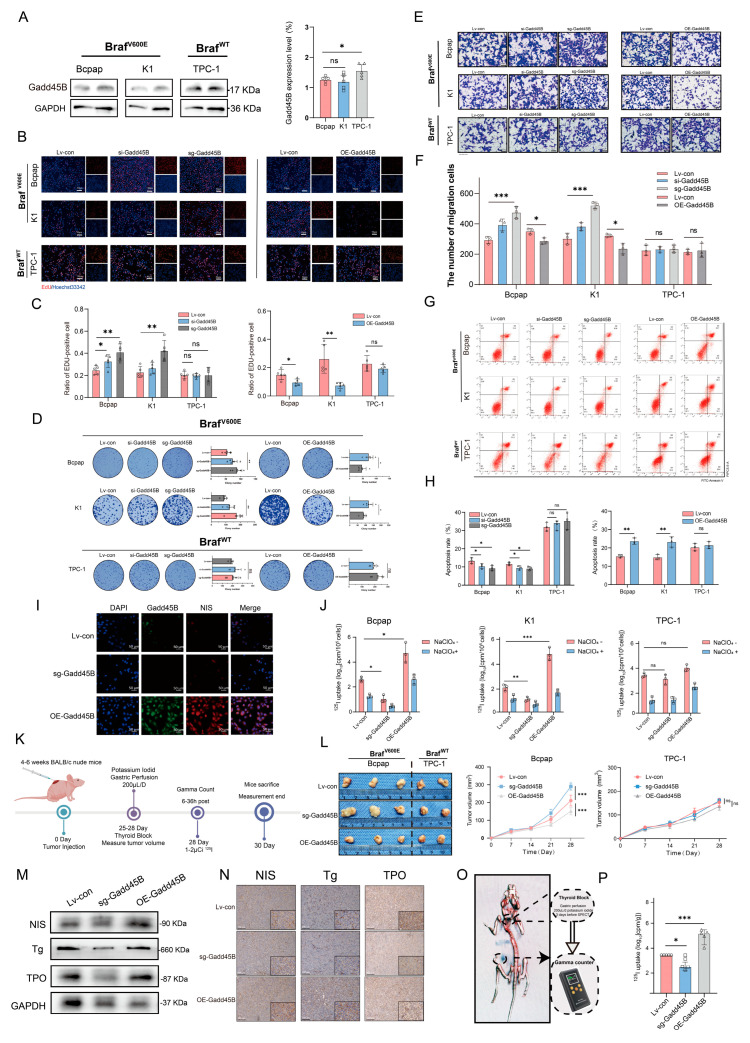
Gadd45B deficiency enhance tumor growth and decrease radioiodine uptake in BRAF^V600E^−mutated thyroid cancer. (**A**) Protein expression levels and quantitative analysis of Gadd45B in Bcpap, K1, and TPC-1 cell lines (n = 5 independent biological replicates). (**B**) Representative images and (**C**) quantitative analysis of EdU assays in three thyroid cancer cell lines with Gadd45B overexpression or downregulation (n = 5 independent biological replicates). Scale bar: 50 μm. (**D**) Colony formation assays and corresponding quantitative analysis of Bcpap, K1, and TPC-1 cell lines with Gadd45B overexpression or downregulation (n = 5 independent biological replicates). (**E**) Transwell migration assays and (**F**) corresponding quantitative analysis of Bcpap, K1, and TPC-1 cell lines with Gadd45B overexpression or downregulation (n = 3 independent biological replicates). Scale bar: 50 μm. (**G**) Apoptosis rates and (**H**) quantitative analysis of Bcpap, K1, and TPC-1 cell lines following Gadd45B overexpression or downregulation, analyzed by flow cytometry using Annexin V−FITC and PI staining (n = 3 independent biological replicates). (**I**) Representative immunofluorescence images showing NIS expression following Gadd45B overexpression or downregulation. Scale bar: 50 μm. (**J**) RAI uptake in the indicated cell lines treated with 0.1% DMSO or 5 mM NaClO_4_ (n = 3 independent biological replicates). (**K**) Workflow of mouse model establishment and assessments (n = 10/group; n = 5 for IHC, Western blotting, survival surveillance, and iodine uptake rate). (**L**) Photographs and tumor volume measurements of harvested tumors from Gadd45B-overexpressed and downregulated CDX models. (**M**) Protein expression levels of NIS, Tg, and TPO in tumors from Gadd45B−overexpressed and downregulated mice. (**N**) Representative immunohistochemistry images showing NIS, Tg, and TPO expression. Scale bar: 100 μm. (**O**) Workflow for radioactivity level surveillance in tumor models. (**P**) Relative radioactivity analysis of tumors from Gadd45B-overexpressed and downregulated tumor−bearing mice. Data are shown as mean ± SD. Statistical significance: ns: not significant, * *p* < 0.05, ** *p* < 0.01, *** *p* < 0.001. The uncropped blots are shown in Appendix A.

**Figure 3 cancers-17-03201-f003:**
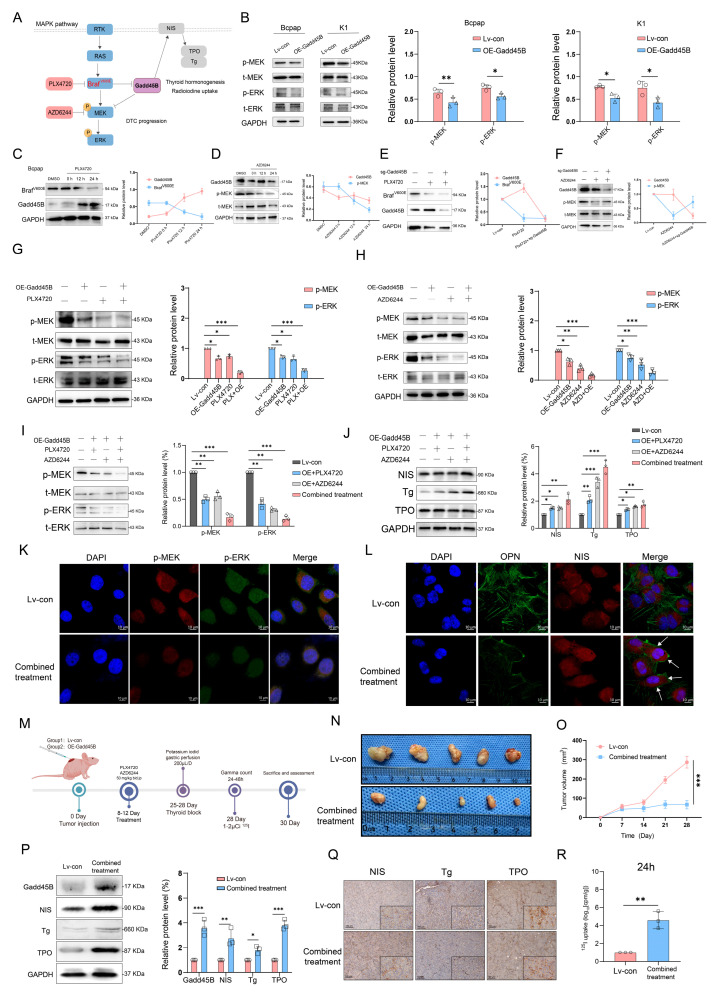
Gadd45B demotes downstream targets of the MAPK pathway and promotes expression of iodine metabolic genes. (**A**) Schematic illustration of canonical targets related to Gadd45B in the MAPK signaling pathway and its effects on NIS, Tg, and TPO expression. (**B**) Protein expression levels and quantitative analysis of key MAPK pathway genes following Gadd45B overexpression in Bcpap and K1 cell lines (n = 3 independent biological replicates). (**C**) Gadd45B and BRAFV600E expression levels in Bcpap cells treated with 0.5 μM PLX4720 for 0, 12, and 24 h, with quantitative analysis (n = 3 independent biological replicates). (**D**) Expression levels of Gadd45B and p-MEK in Bcpap cells treated with 1 μM AZD6244 for the indicated durations, with quantitative analysis (n = 3 independent biological replicates). (**E**) Effects of Gadd45B depletion alone or in combination with 0.5 μM PLX4720 on indicated protein expression levels, with quantitative analysis (n = 3 independent biological replicates). (**F**) Effects of Gadd45B overexpression alone or in combination with 1 μM AZD6244 on protein expression levels, with quantitative analysis (n = 3 independent biological replicates). (**G**) Western blot analysis showing the effects of Gadd45B overexpression and/or 0.5 μM PLX4720 on p-MEK, t-MEK, p-ERK, and t-ERK expression levels in Bcpap cells, with quantitative analysis (n = 3 independent biological replicates). (**H**) Western blot analysis showing the effects of Gadd45B overexpression and/or MEK1/2 inhibition (1 μM AZD6244) on p-MEK, t-MEK, p-ERK, and t-ERK expression levels in Bcpap cells, with quantitative analysis (n = 3 independent biological replicates). (**I**) Western blot analysis showing expression levels of p-MEK, t-MEK, p-ERK, and t-ERK in different treatment groups in Bcpap cells. (**J**) Western blot analysis showing expression levels of NIS, Tg, and TPO under control, Gadd45B overexpression, PLX4720 treatment, and AZD6244 treatment, with quantitative analysis. (**K**) Representative immunofluorescence images showing the subcellular localization of p-MEK and p-ERK. (**L**) Immunofluorescence images showing OPN and NIS localization in Bcpap cells following combined treatment with Gadd45B overexpression, BRAFV600E inhibition, and MEK1/2 inhibition. Scale bar: 10 μm. (**M**) Workflow for mouse model establishment and assessment (n = 6/group; n = 3 for IHC, Western blotting, and tumor growth surveillance; n = 3 for iodine uptake rate). (**N**) Representative images of tumors from control and combined treatment groups. (**O**) Tumor volume was monitored weekly in indicated groups. (**P**) Protein expression levels of Gadd45B, NIS, Tg, and TPO in tumors from indicated groups, with quantitative analysis. (**Q**) Immunohistochemistry analysis of Gadd45B, NIS, Tg, and TPO expression levels in tumors from indicated groups. (**R**) Quantitative analysis of RAI uptake levels at 24 h in vivo. Data are shown as mean ± SD. Statistical significance: ns: not significant, * *p* < 0.05, ** *p* < 0.01, *** *p* < 0.001. The uncropped blots are shown in Appendix A.

**Figure 4 cancers-17-03201-f004:**
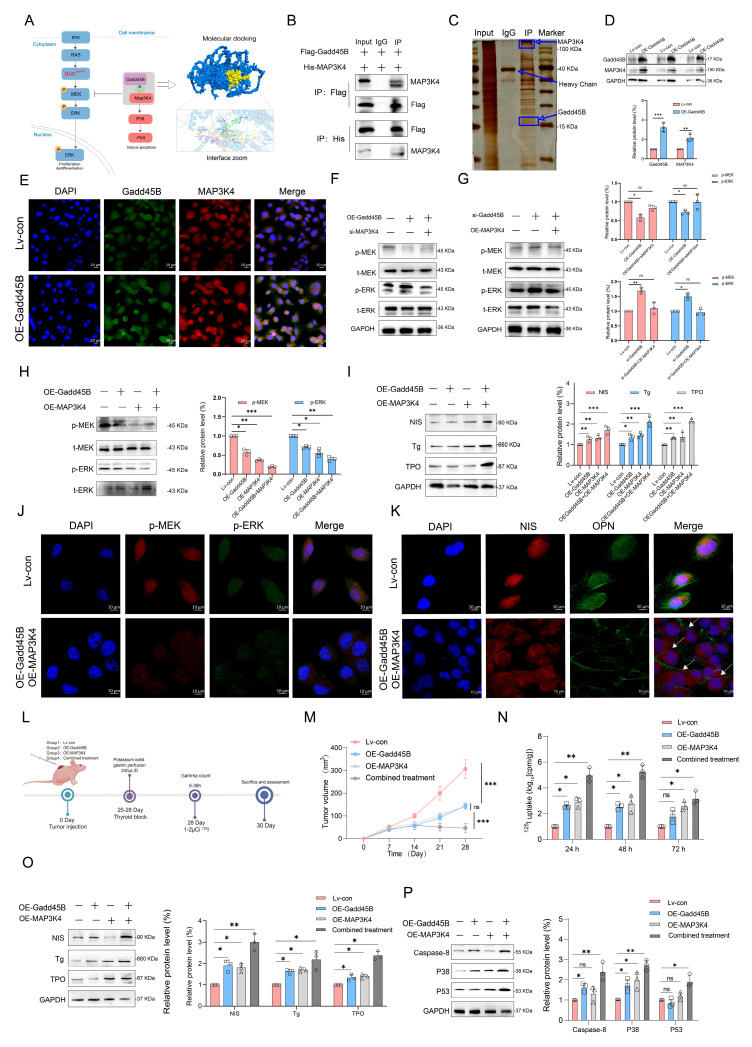
The synergistic relationship of Gadd45B and MAP3K4 in the MAPK pathway enhances antitumor activity. (**A**) Schematic representation of the biological processes involving Gadd45B and MAP3K4, with details of their molecular docking analysis. (**B**) Co-immunoprecipitation (Co-IP) assays showing the interaction between Gadd45B and MAP3K4 in Bcpap (OE-Gadd45B) cells (n = 3 independent biological replicates). (**C**) Silver staining assays confirming the interaction between Gadd45B and MAP3K4. (**D**) Effects of Gadd45B overexpression on MAP3K4 protein expression levels, with quantitative analysis (n = 3 independent biological replicates). (**E**) Representative immunofluorescence images showing the subcellular localization of Gadd45B and MAP3K4. Scale bar: 10 μm. (**F**) Western blot analysis showing the effects of Gadd45B overexpression and MAP3K4 knockdown on the MAPK signaling pathway, with quantitative analysis (n = 3 independent biological replicates). (**G**) Western blot analysis showing the effects of Gadd45B knockdown and MAP3K4 overexpression on the MAPK signaling pathway, with quantitative analysis (n = 3 independent biological replicates). (**H**) Synergistic effects of Gadd45B and MAP3K4 overexpression on the MAPK signaling pathway, with quantitative analysis (n = 3 independent biological replicates). (**I**) Synergistic effects of Gadd45B and MAP3K4 overexpression on the expression levels of NIS, Tg, and TPO, with quantitative analysis (n = 3 independent biological replicates). (**J**) Representative immunofluorescence images showing the subcellular localization and expression levels of p-MEK and p-ERK. (**K**) Immunofluorescence images depicting NIS localization and expression in Bcpap cells following combined Gadd45B and MAP3K4 overexpression. Scale bar: 10 μm. (**L**) Schematic workflow for mouse model establishment and assessment, with groups: control, Gadd45B overexpression, MAP3K4 overexpression, and combined treatment (Gadd45B and MAP3K4 overexpression). n = 6 per group (n = 3 for IHC, Western blotting, and tumor growth surveillance; n = 3 for iodine uptake rate). (**M**) Tumor volume curves for the indicated treatment groups. (**N**) Quantitative analysis of RAI uptake levels at 24, 48, and 72 h in the indicated treatment groups. (**O**) Western blot analysis showing the synergistic effects of Gadd45B and MAP3K4 overexpression on the expression levels of NIS, Tg, and TPO. (**P**) Western blot analysis showing the expression of caspase-8, p38, and p53 proteins in the combined treatment group, with corresponding quantitative analysis. Data are shown as mean ± SD. Statistical significance: ns: not significant, * *p* < 0.05, ** *p* < 0.01, *** *p* < 0.001. The uncropped blots are shown in Appendix A.

**Figure 5 cancers-17-03201-f005:**
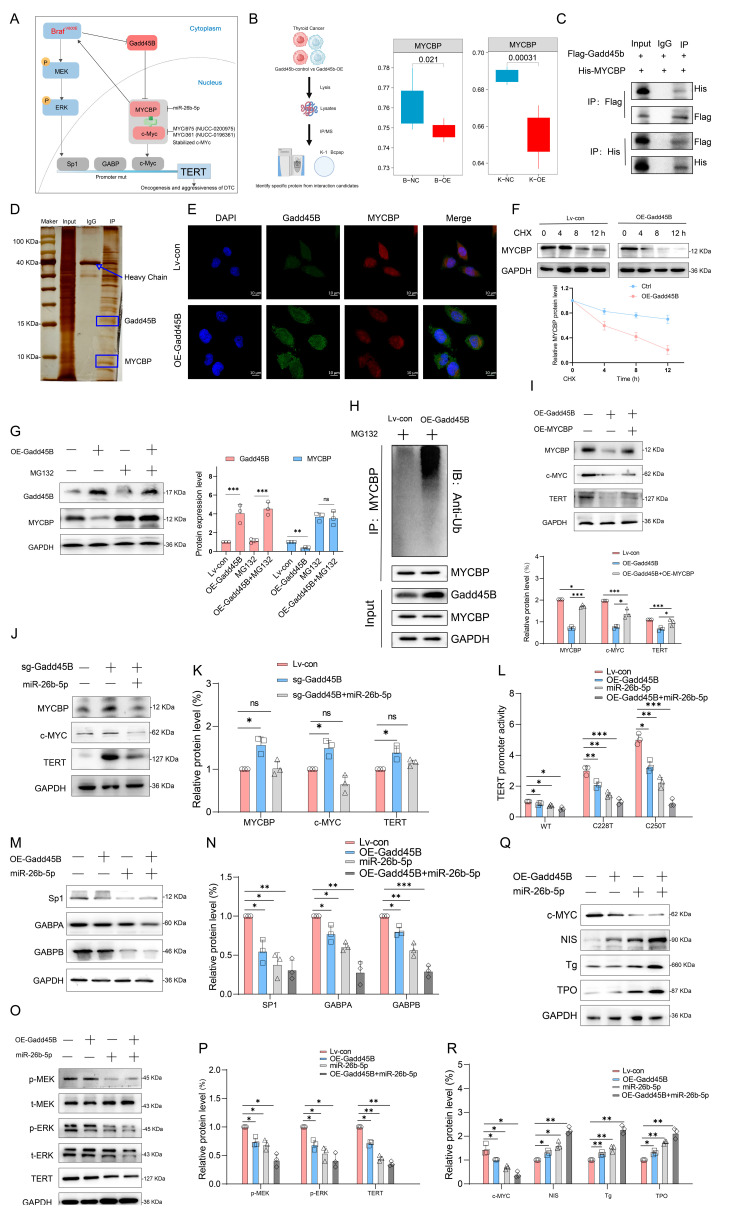
Deficiency of Gadd45B enhances the stability of the MYCBP-c-MYC complex and TERT promoter activity by alleviating MYCBP degradation stress. (**A**) Schematic illustration of the effects of Gadd45B on the MYCBP-c-MYC complex and TERT promoter activity. (**B**) Workflow and analysis of immunoprecipitation-mass spectrometry (IP-MS) for Gadd45B in BRAFV600E-mutated cell lines. (**C**) Co-immunoprecipitation (Co-IP) and Western blot (WB) analysis of Gadd45B and MYCBP in Bcpap cells (n = 3 independent biological replicates). (**D**) Silver staining assay validating the interaction between Gadd45B and MYCBP in Bcpap cells (n = 3 independent biological replicates). (**E**) Representative immunofluorescence images showing the subcellular localization and expression levels of Gadd45B and MYCBP (scale bar: 10 µm; n = 3 independent biological replicates). (**F**) MYCBP expression levels in Bcpap cells treated with 1 µM MG(CHX) for the indicated durations, with relative quantification (n = 3 independent biological replicates). (**G**) Effects of MG132 treatment on MYCBP expression levels in Bcpap cells, with quantitative analysis (n = 3 independent biological replicates). (**H**) Ubiquitination levels of MYCBP in Bcpap cells with Gadd45B overexpression (n = 3 independent biological replicates). (**I**) Protein expression levels of MYCBP, c-MYC, and TERT in Bcpap cells with Gadd45B overexpression, alone or combined with MYCBP overexpression, with quantitative analysis (n = 3 independent biological replicates). (**J**,**K**) Protein expression levels of MYCBP, c-MYC, and TERT in Bcpap cells with Gadd45B knockout, alone or combined with MYCBP downregulation (50 nM miR-26b-5p transfection), and corresponding quantitative analysis (n = 3 independent biological replicates). (**L**) Dual-luciferase reporter assays for wild-type, C228T, and C250T-mutated TERT promoters under the following conditions: control, Gadd45B overexpression, miR-26b-5p, and their combination (n = 3 independent biological replicates). (**M**,**N**) Protein expression levels of Sp1, GABPA, and GABPB, and their quantification in the indicated groups (n = 3 independent biological replicates). (**O**,**P**) Degree of MAPK signaling pathway activation in the indicated groups, with quantitative analysis (n = 3 independent biological replicates). (**Q**,**R**) Protein expression levels of c-MYC and iodine metabolism proteins (NIS, Tg, and TPO), with quantitative analysis (n = 3 independent biological replicates). Data are shown as mean ± SD. Statistical significance: ns: not significant, * *p* < 0.05, ** *p* < 0.01, *** *p* < 0.001. The uncropped blots are shown in Appendix A.

**Figure 6 cancers-17-03201-f006:**
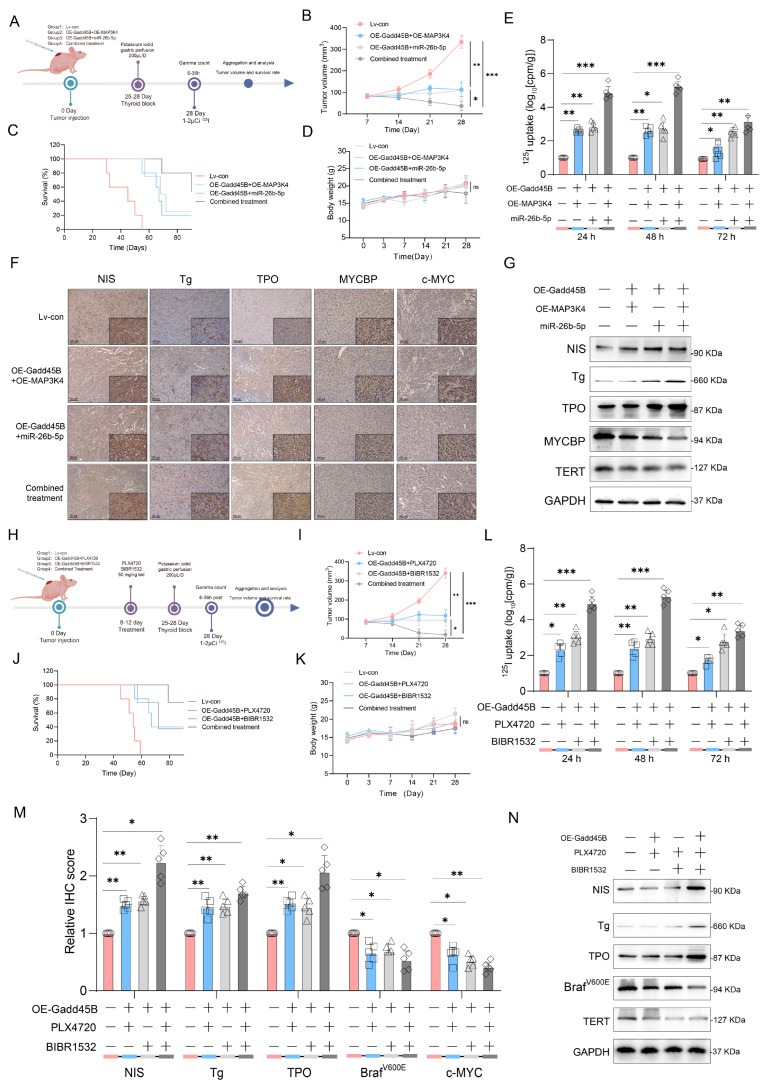
Gadd45B overexpression and targeted inhibitors drive the restoration of survival rate and iodine metabolic genes of thyroid cancer burden mouse models. (**A**) Schematic of the experimental workflow for iodine uptake rate assessment and tumor burden surveillance in Bcpap tumor-bearing mice. Groups: Lv-control, Gadd45B and MAP3K4 overexpression, Gadd45B downregulation and miR-26b-5p inhibition, Gadd45B and MAP3K4 overexpression combined with miR-26b-5p inhibition (n = 10/group; for IHC, Western blotting, and survival surveillance: n = 5; for iodine uptake rate: n = 5). (**B**) Tumor volume measurements over time for the indicated groups. (**C**) Survival rate analysis of mice treated with the indicated conditions. (**D**) Body weight curves of mice in each treatment group. (**E**) Iodine uptake rate for the indicated groups, n = 5. (**F**) Representative IHC images of NIS, Tg, TPO, MYCBP, and c-MYC expression in tumors from each group (scale bar: 100 µm). (**G**) Protein expression levels of NIS, Tg, TPO, MYCBP, and TERT in the indicated groups. (**H**) Schematic of the second workflow involving iodine uptake rate assessment with PLX4720 and BIBR1532 treatments in Bcpap tumor-bearing mice. Groups: Lv-control, Gadd45B overexpression + 1 μM PLX4720, Gadd45B overexpression + 20 μM BIBR1532, and combined treatment (Gadd45B overexpression + PLX4720 + BIBR1532) (n = 10/group; for IHC, Western blotting, and survival surveillance: n = 5; for iodine uptake rate: n = 5). (**I**) Tumor volume measurements over time for mice treated under the indicated conditions. (**J**) Survival rate analysis for mice in different treatment groups. (**K**) Body weight trends across treatment groups. (**L**) Iodine uptake rate for the indicated treatment groups. (**M**) Quantitative analysis of IHC images for NIS, Tg, TPO, BRAFV600E, and c-MYC expression in tumors from the indicated groups. (**N**) Protein expression levels of NIS, Tg, TPO, BRAFV600E, and TERT across groups, with quantitative analysis. Data are shown as mean ± SD. Statistical significance: ns: not significant, * *p* < 0.05, ** *p* < 0.01, *** *p* < 0.001. The uncropped blots are shown in Appendix A.

**Figure 7 cancers-17-03201-f007:**
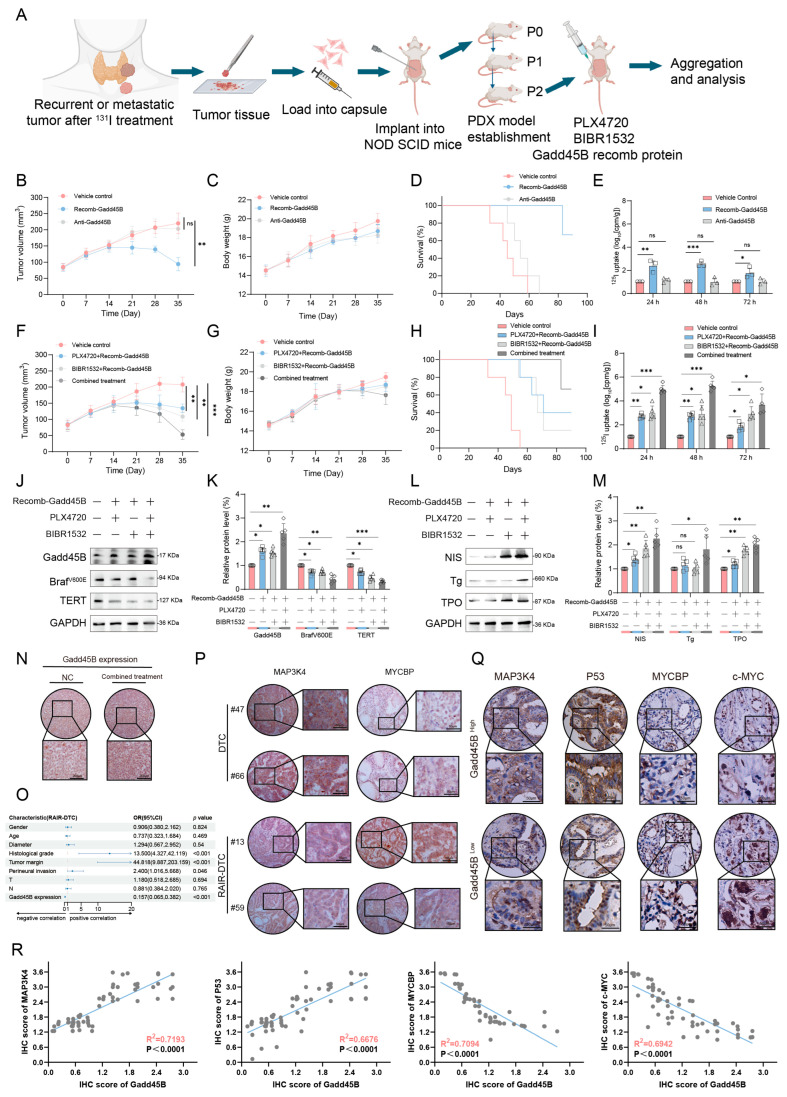
Recombinant Gadd45B protein enhance uptake of radioiodine and reverse dedifferentiation in PDX models of RAIR-DTC. (**A**) Schematic workflow for establishing patient-derived xenograft (PDX) models of RAIR-DTC. (n = 3 for panels B–E, and n = 5 for panels F–M). (**B**) Tumor volume measurements over time for the following groups: control, recombinant Gadd45B protein, and anti-Gadd45B antibody groups. (**C**) Body weight curves for mice in the indicated groups, assessed weekly. (**D**) Survival rate analysis of mice treated under the indicated conditions. (**E**) Relative radioactivity (iodine uptake) analysis after 125I perfusion at 24h, 48h, and 72h across the indicated groups. (**F**) Tumor volume measurements over time for the following groups: control, PLX4720 + recombinant Gadd45B protein, BIBR1532 + recombinant Gadd45B protein, and the combined treatment group (PLX4720, BIBR1532, and recombinant Gadd45B protein). (**G**) Body weight measurements for the indicated groups, assessed weekly. (**H**) Survival rate analysis of mice under the different treatment conditions. (**I**) Relative radioactivity analysis after 125I perfusion at 24h, 48h, and 72h across treatment groups. (**J**) Western blot analysis of BRAFV600E and TERT protein expression levels in tumors from the indicated groups, with (**K**) quantitative analysis. (**L**) Western blot analysis of NIS, Tg, and TPO protein expression levels in tumors from the indicated groups, with (**M**) quantitative analysis. (**N**) Representative IHC staining images of Gadd45B in tumor tissues (scale bar: 200 µm). (**O**) Forest plot showing multivariate analysis results for factors associated with RAIR-DTC from cohort 4. Odds ratio (OR): 0.157, 95% confidence interval (CI): 0.065–0.382, *p* < 0.001. (**P**) Representative IHC staining images of MAP3K4 and MYCBP from cohort 4 (scale bar: 50 µm). (**Q**) Representative IHC staining images of MAP3K4, P53, MYCBP, and c-MYC in cohort 4, grouped by Gadd45B expression levels (scale bar: 50 µm). (**R**) Correlation analysis between Gadd45B expression and MAP3K4, P53, MYCBP, or c-MYC expression in cohort 4. Data are shown as mean ± SD. Statistical significance: ns: not significant, * *p* < 0.05, ** *p* < 0.01, *** *p* < 0.001. The uncropped blots are shown in Appendix A.

## Data Availability

Data supporting the reported results are contained within the article and its Appendix A. Additional data are unavailable due to privacy and ethical restrictions.

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
