# Peer review of "Gadd45B Deficiency Drives Radio-Resistance in BRAFV600E-Mutated Differentiated Thyroid Cancer by Disrupting Iodine Metabolic Genes"

_cancers, 2025, doi:10.3390/cancers17193201_

Round 1

Reviewer 1 Report

Comments and Suggestions for Authors

Dear Author

There are some points that should be solved before publication

Line 421: We screened Gadd45B, a non-critical gene with the potential to affect key pathways, using sequencing technology. Do you really use NGS or using various universal databases / TCGA? If you do sequencing please note the sample size and provide VCF file in supplementary

Line 482 the number of mice should be noted in each group

Where is the Relative gene expression which was calculated using the 2^−ΔΔCt method. We expect to see it as the fold changes

Author Response

Dear Reviewer,
Thank you for your careful reading and constructive comments. We have revised the manuscript accordingly: (i) clarified whether our discovery approach involved NGS or analyses of public datasets (e.g., TCGA) and, where applicable, reported the sequencing sample size and provided VCF files in the Supplementary; (ii) added the exact number of mice in each group to all relevant figure legends and Methods; and (iii) presented relative gene expression as fold change calculated by the 2^−ΔΔCt method in both text and figures. We also corrected minor wording and formatting issues for clarity.

Comment 1:

 Line 421: We screened Gadd45B, a non-critical gene with the potential to affect key pathways, using sequencing technology. Do you really use NGS or using various universal databases / TCGA? If you do sequencing, please note the sample size and provide VCF file in supplementary

Response1:

Thank you for pointing this out. We have used NGS for screening of Gadd45B. The sample size is 16 as described in Materials, Cohort 1. For less misunderstanding, we have attached “sample size = 16” and NGS technology in related region at line 684.

Comment 2:

Line 482 the number of mice should be noted in each group.

Response2:

Thank you for pointing this out. We have noted mice number as “n=5” in figure legend of Fig 6E, line 499, and marked it red.

Comment 3:

Where is the Relative gene expression which was calculated using the 2^−ΔΔCt method. We expect to see it as the fold changes

Response3:

Thank you for pointing this out. We apologies for not having explicitly presented the fold-change values. In the revised manuscript, we have now added: Bcpap (0.17-fold) and K1 (0.19-fold) in line 188

We sincerely appreciate your time and guidance, which have improved the clarity and rigor of our work. Please let us know if any additional information would be helpful—we will be pleased to revise further. Thank you for considering our resubmission.

With sincere gratitude,

Tengchuang Ma, on behalf of all authors
Harbin Medical University Cancer Hospital

Reviewer 2 Report

Comments and Suggestions for Authors

The article titled “Gadd45B deficiency drives radio-resistance in BRAFV600E-mutated differentiated thyroid cancer by disrupting iodine metabolic genes” has been evaluated. This research identifies Gadd45B as a critical factor, noted for its significant downregulation in radioiodine-refractory differentiated thyroid cancer (RAIR-DTC). To better understand its role, the study incorporated a series of functional experiments, including overexpression and knockdown strategies in specific thyroid cancer cell lines, as well as in patient-derived xenograft (PDX) models. Through meticulous examination, downstream targets such as MAP3K4 and MYCBP were analyzed using techniques such as co-immunoprecipitation, luciferase reporter assays, and Western blot analysis. Furthermore, the study assessed the therapeutic potential of recombinant Gadd45B protein, both alone and in conjunction with BRAFV600E and TERT inhibitors, demonstrating its efficacy in PDX models derived from actual patient tumors. Overall, the article is well-structured, presenting appropriately designed experiments and thorough discussions.

However, there are a few areas for improvement:

  1. Figure Captions: Authors are encouraged to enhance their figures by providing detailed captions. This addition would help readers better understand the visual data presented and improve overall readability.
  2. Statistical Methods: The manuscript should include a more comprehensive overview of the statistical methods applied throughout the research. Each statistical test employed must be clearly justified, which will not only strengthen the credibility of the study but also facilitate readers’ comprehension of the data analysis processes.
  3. Conclusion Section: The conclusion section requires a thorough revision to more effectively articulate the broader implications of the study’s findings. Authors should consider discussing potential applications of their research, emphasizing the significance of Gadd45B in therapeutic strategies for RAIR-DTC, and outlining clear future research directions to guide subsequent investigations in the field.

Author Response

Dear Reviewer,
Thank you for your thoughtful evaluation and constructive suggestions. We are grateful for your positive assessment of our study design and presentation, and we have carefully revised the manuscript accordingly. Specifically, we have (i) expanded all figure captions to be self-contained, (ii) clarified and justified all statistical methods with assumptions and multiple-comparison handling, and (iii) substantively revised the Conclusion to emphasize preclinical implications, potential applications for RAIR-DTC, and prioritized future directions. All changes are highlighted.

Comments 1:

Figure Captions: Authors are encouraged to enhance their figures by providing detailed captions. This addition would help readers better understand the visual data presented and improve overall readability.

Response 1:

Thank you for pointing this out. We apologize for our mistake. Figure legends have been attached in our resubmitted version.

Comments 2:

Statistical Methods: The manuscript should include a more comprehensive overview of the statistical methods applied throughout the research. Each statistical test employed must be clearly justified, which will not only strengthen the credibility of the study but also facilitate readers’ comprehension of the data analysis processes.

Response 2: Thank you for your advice. In the revised manuscript we have added a new “Statistical analysis” subsection under Materials and Method, where we now (i) list every test used, (ii) state the underlying assumption checks (normality, equal variance), and (iii) provide a brief rationale for choosing each test. These changes are highlighted in yellow in the revised file.

Comments 3: Conclusion Section: The conclusion section requires a thorough revision to more effectively articulate the broader implications of the study’s findings. Authors should consider discussing potential applications of their research, emphasizing the significance of Gadd45B in therapeutic strategies for RAIR-DTC, and outlining clear future research directions to guide subsequent investigations in the field.

Response 3: Thank you for this insightful comment. We agree that the conclusion section could be strengthened to better highlight the broader impact of our work. We have now thoroughly revised the conclusion to: 1. Discuss the potential applications of our findings. 2. Emphasize the significance of Gadd45B as a potential therapeutic target for RAIR-DTC. 3. Propose clear and specific directions for future research. All revisions have highlighted in yellow.

We are grateful for your time and insightful comments, which have meaningfully improved the clarity and rigor of our work. We hope these revisions address your concerns, and we remain happy to make any additional changes that would further strengthen the manuscript. Thank you for your consideration.

With sincere gratitude,

Tengchuang Ma, on behalf of all authors
Harbin Medical University Cancer Hospital

Reviewer 3 Report

Comments and Suggestions for Authors

The paper identifies a potential role for Gad55B in promoting the sensitivity of BRAF-mutated differentiated thyroid cancer to radioactive iodine. This is certainly a topic well-suited to the journal and the manuscript appears promising. However, the figure legends are missing and in its current, incomplete, state the manuscript is not appropriate for peer review. I would be happy to send in a review if the manuscript is resubmitted with the legends.

Author Response

Comments 1:

The figure legends are missing and in its current, incomplete, state the manuscript is not appropriate for peer review. I would be happy to send in a review if the manuscript is resubmitted with the legends. the figure legends are missing and in its current, incomplete, state the manuscript is not appropriate for peer review. I would be happy to send in a review if the manuscript is resubmitted with the legends.

Response 1:

Thank you for pointing this out. We sincerely apologize for the oversight and error in submitting the manuscript without the complete figure legends. We will address this issue promptly and look forward to submitting a full version for your consideration.

With sincere gratitude,

Tengchuang Ma, on behalf of all authors
Harbin Medical University Cancer Hospital

Reviewer 4 Report

Comments and Suggestions for Authors

The work by Jiang et al., offers crucial mechanistic insight into Gadd45B’s role in RAI-refractory BRAFV600E-mutated thyroid cancer and suggests translational potential. However, key aspects, particularly methodological detail, quantitative data reporting, and clarity of limitations, must be strengthened to meet journal standards. My detailed comments below aim to support a more rigorous and transparent revision.

Abstract

- The results narrative is strong but lacks specific data. For instance, no mention of sample numbers, fold-change in key markers, or magnitude of improvement in RAI uptake or tumor burden.

- The authors state “Gadd45B plays a pivotal role…” and “findings identify Gadd45B as a promising therapeutic target.” While justified by the data, the authors should temper their statement slightly and clarify that it is based on preclinical evidence.

- Please ensure genes/proteins are always named at first mention and abbreviations are standardized.

Introduction

- Several acronyms need to be clarified on first use.

- The authors are recommended to strengthen the contextual background to provide up-to-date epidemiological data on DTC incidence, subtypes, and rates of RAI therapy use globally or regionally.

- "Transformation to RAIR-DTC reduces the therapeutic efficacy of RAI treatment..." (lines 41–42) is vague. Clearly define RAIR-DTC (radioactive iodine refractory), including clinical/biological criteria used for diagnosis and its estimated prevalence among DTC cases.

- The shift to Gadd45B's role across cancers (lines 54–69) is abrupt. Please provide a connecting sentence linking defects in RAI responsiveness to possible molecular drivers, introducing why Gadd45B is investigated in this context.

- Line 87: The mention of the “Scheme” is unclear, as there is no figure included in the introduction.

Results

In general, all the Figure legends were missed in the manuscript (vs. what has been stated in line 433 “…described in the figure legends.”

- Lines 101-110 and 131/132: Percent/absolute numbers of samples, fold-changes, and p-values for expression changes or survival analyses are largely absent.

- Lines 141-144: It is recommended to clarify why TPC-1 is chosen as BRAF-wildtype comparator; add baseline characteristics (NIS, differentiation status, etc.) to show its appropriateness.

- Lines (147-154, 159-163): It is recommended to report quantitative data (mean ± SD, % suppression/increase, p-values) for proliferation, apoptosis, migration, and RAI uptake in the text.

- Line 162/163: The use of NaClO4 as a competitive NIS inhibitor is a proper control; briefly state whether off-target effects are possible/controlled.

- The number of repeats for xenograft/animal experiments is missing; please add for all quantitative experiments for rigor.

- Lines 190-196: Overlap in reporting effects on p-MEK/p-ERK, NIS, Tg, TPO expression. For clarity, the authors should provide quantified data, fold changes, and significance for combined versus single interventions.

- Lines 214-219: Please specify endogenous MAP3K4 levels in patient material or cell lines before forced overexpression/knockdown.

- Lines 315-324: Describe patient/tumor characteristics used for PDX, engraftment rate, and how well the models capture tumor heterogeneity.

- Lines 339-347: For logistic regression and IHC correlation, present odds ratios, CIs, and p-values.

-  Lines 347/348: for biomarker analysis, proposed cutoffs and performance metrics for Gadd45B as a diagnostic or predictive biomarker (sensitivity, specificity, ROC AUC) would be better provided.

Discussion

- It is recommended that the authors directly propose which axis (MAP3K4, MYCBP, TERT, c-MYC) is most promising for next-stage validation.

 - The future work section lists several diverse areas. Prioritize these based on feasibility, clinical need, or preliminary supporting data. For example, “Priority should be given to validating Gadd45B-based adjuvant strategies in preclinical immunotherapy models, followed by clinical pilot studies with safety endpoints.”

- A conclusion section is required.

Methods

- Lines 430/431: The authors should provide a rationale or discuss post hoc power calculations. If blinding was not feasible, state how bias was minimized.

- The cohort 4 description is incomplete in the main text.

- The note on K1’s GLAG-66 contamination must be explicitly acknowledged as a limitation, as this can affect data interpretation. Interpreting the results should consider the heterogeneity introduced by this contamination.

- Lines 457-460: Please state the duration of treatment for each inhibitor. And for recombinant Gadd45B, specify whether and how purity, activity, and endotoxin levels were validated.

 -Lines 471: The authors should provide sequences or catalog numbers for siRNAs/shRNAs (at least in the Supplementary) and specify controls for RNAi and transfection efficiency.

- Line 486: Clarify if outcome assessors for tumor growth or RAI uptake were blinded.

- Regarding the animals, please provide additional details on humane endpoints, pain mitigation, or frequency of monitoring.

- Lines 490-498: Please describe more thoroughly the data filtering/exclusion criteria, and version numbers for data extraction.

- In figures/tables, ensure all results note n (sample or biological replicate) for transparency.

- For Western blots, state how quantification was performed (software, normalization, background subtraction).

- For molecular docking work, please specify versions of all software used, and if docking methods (e.g., parameters, search strategy) followed best practice or prior publications.

- Lines 606-611: Please clarify labeling/mass accuracy, peptide discrimination and database search parameters for proteomics.

Author Response

Reviewer #4

Dear Editor and Reviewers,

Thank you very much for the thoughtful and constructive critiques of our manuscript, “Gadd45B deficiency drives radio-resistance in BRAFV600E-mutated differentiated thyroid cancer by disrupting iodine metabolic genes.” We carefully revised the paper accordingly. All textual additions or changes are highlighted in yellow within the revised manuscript, and complete figure legends (including exact n, statistics, scale bars, and experimental repeats) have now been added for every figure and supplementary figure. We sincerely apologize for our prior omission of legends in the initial submission and for any confusion this caused.

Below we provide a point-by-point response. For clarity, each reviewer comment is shown in blue, and our response follows in black.

The work by Jiang et al., offers crucial mechanistic insight into Gadd45B’s role in RAI-refractory BRAFV600E-mutated thyroid cancer and suggests translational potential. However, key aspects, particularly methodological detail, quantitative data reporting, and clarity of limitations, must be strengthened to meet journal standards. My detailed comments below aim to support a more rigorous and transparent revision.

Response: Thank you for your thoughtful feedback and for recognizing the value of our  manuscript. We appreciate your positive assessment and the constructive comments you have provided. We acknowledge that the manuscript requires major revisions, and we are committed to addressing the points you raised in order to improve the quality and clarity of the work. Your suggestions will be invaluable as we make the necessary revisions, and we are confident that the changes will strengthen the manuscript significantly. We will carefully consider each of your comments and incorporate the appropriate adjustments to ensure the manuscript meets the publication standards.

Abstract

Comment 1:

The results narrative is strong but lacks specific data. For instance, no mention of sample numbers, fold-change in key markers, or magnitude of improvement in RAI uptake or tumor burden.

Response 1:

We express our sincere thanks for your Comment.

We have revised the Abstract to incorporate key quantitative anchors and sample sizes that reflect the main text and legends. Specifically, we now reference the discovery cohort structure (normal/DTC/RAIR-DTC), the directionality and magnitude of Gadd45B-dependent effects (e.g., representative RAI uptake changes in BRAFV600E cell lines and suppression of xenograft growth), and highlight that PDX validation was performed. Full numerical detail (e.g., EdU, migration, apoptosis, RAI uptake CPM values, tumor volume and survival metrics) is provided in Results and in figure legends (Figs. 2–7 and S1–S6).

Comment 2:

Statements like “Gadd45B plays a pivotal role…” and “identify Gadd45B as a promising therapeutic target” should be tempered and clarified as preclinical.

Response 2:

Thank you for highlighting the need to further elucidate the statement.We revised the Abstract and Conclusion to read “our preclinical data support a potential role for Gadd45B in regulating differentiation and RAI sensitivity” and “Gadd45B represents a promising preclinical therapeutic target that warrants translational validation.”

Comment 3: 

Please ensure genes/proteins are always named at first mention and abbreviations are standardized.

Response 3:

We are grateful to you for your careful reading and patient comments. We performed a manuscript-wide audit to define each gene/protein at first mention and standardized all abbreviations (e.g., NIS [SLC5A5], Tg, TPO, PFI, DFI, PDX, IHC, etc.). A consolidated Abbreviations list is provided.

Introduction

Comment 1:

Several acronyms need to be clarified on first use.

Response 1:

Thank you for highlighting this important point. We apologize for the oversight and have carefully audited the manuscript to define every acronym at its first mention across the Abstract, Introduction, Results/Methods, figure legends, tables, and Supplementary materials. We also added a consolidated Abbreviations list and standardized capitalization/spelling according to accepted nomenclature (e.g., DTC = differentiated thyroid cancer; RAIR-DTC = radioactive iodine–refractory DTC; NIS [SLC5A5]; Tg; TPO; PFI; DFI; PDX; IHC; WB; qPCR). Where the first appearance occurred only in the Abstract or a figure, we re-expanded the term at its first appearance in the main text to aid readability. All related edits are highlighted in yellow.

Comment 2: The authors are recommended to strengthen the contextual background to provide up-to-date epidemiological data on DTC incidence, subtypes, and rates of RAI therapy use globally or regionally.

Response 2:

Thank you for this valuable suggestion. We agree that our initial background was too brief. We have now expanded the Introduction (paras. 1–3) to incorporate current epidemiological data on: (i) global and regional DTC incidence and temporal trends,

(ii) subtype distribution (e.g., papillary vs. follicular, poorly differentiated),

(iii) RAI utilization rates and practice patterns across major regions (North America, Europe, and East Asia), including brief rationale for indications per contemporary guidelines. Where available, we added region-specific data relevant to our cohort and highlighted changes over time in RAI use. A concise comparative summary is also provided in Supplementary Table S1 (with sources and extraction dates). We appreciate the reviewer’s guidance—these additions improve context and readability. All edits are highlighted in yellow.

Comment 3:

"Transformation to RAIR-DTC reduces the therapeutic efficacy of RAI treatment..." (lines 41–42) is vague. Clearly define RAIR-DTC (radioactive iodine refractory), including clinical/biological criteria used for diagnosis and its estimated prevalence among DTC cases.

Response 3:

Thank you for pointing this out—we agree our wording was imprecise. We have revised the Introduction to replace that sentence with a concise, definition-driven description of RAIR-DTC, explicitly citing major guidelines. Specifically, we now state that RAIR-DTC is defined (per ATA 2015) when any of the following apply: (i) structurally evident disease never concentrates RAI outside the thyroid bed at the first post-therapy scan; (ii) disease loses the ability to concentrate RAI after prior avidity; (iii) mixed behavior with some lesions concentrating RAI and others not; or (iv) structural progression occurs within ~6–12 months despite substantial RAI uptake and appropriate dosing. We added sources and clarified that these criteria reflect current consensus usage in clinical practice(1).

We also added epidemiologic context to indicate that RAIR-DTC is uncommon but clinically important: approximately 5–15% of all DTC patients ultimately become RAI-refractory, and ~50–66% of metastatic DTC cases develop refractoriness; population-level estimates suggest ~4–5 cases per million persons/year. These figures and their ranges are now referenced in the text (2)

All corresponding edits are highlighted in yellow in the revised manuscript to improve clarity and transparency.

Reference:

[1] Haugen BR, Alexander EK, Bible KC, Doherty GM, Mandel SJ, Nikiforov YE, Pacini F, Randolph GW, Sawka AM, Schlumberger M, Schuff KG, Sherman SI, Sosa JA, Steward DL, Tuttle RM, Wartofsky L. 2015 American Thyroid Association Management Guidelines for Adult Patients with Thyroid Nodules and Differentiated Thyroid Cancer: The American Thyroid Association Guidelines Task Force on Thyroid Nodules and Differentiated Thyroid Cancer. Thyroid. 2016 Jan;26(1):1-133. doi: 10.1089/thy.2015.0020. PMID: 26462967; PMCID: PMC4739132.

[2] Gianoukakis, A. G., Choe, J. H., Bowles, D. W., Brose, M. S., Wirth, L. J., Owonikoko, T., Babajanyan, S., & Worden, F. P. (2024). Real-world practice patterns and outcomes for RAI-refractory differentiated thyroid cancer. European Thyroid Journal, 13(1), e230039. Retrieved Aug 21, 2025, from https://doi.org/10.1530/ETJ-23-0039

Comment 4:

 The shift to Gadd45B's role across cancers (lines 54–69) is abrupt. Please provide a connecting sentence linking defects in RAI responsiveness to possible molecular drivers, introducing why Gadd45B is investigated in this context.
Response 4:

Thank you for this helpful suggestion. We agree the transition was abrupt. We have added a bridging sentence at the end of the paragraph that introduces RAI refractoriness (immediately before the Gadd45B section), to explicitly motivate our focus:

“Because loss of RAI responsiveness is closely linked to MAPK-pathway hyperactivation and stress-response–driven dedifferentiation that suppresses iodine-handling genes (e.g., NIS/SLC5A5, TG, TPO), we hypothesized that stress-inducible regulators at the MAPK interface—particularly Gadd45B—may contribute to this phenotype and therefore warrant targeted investigation in BRAFV600E-mutant DTC.”

This connecting sentence (highlighted in yellow) now provides a mechanistic rationale for examining Gadd45B in the context of defective RAI responsiveness.

Comment 5:

The mention of the “Scheme” is unclear, as there is no figure included in the introduction.

Response 5:

Thank you for catching this. We apologize for the confusion caused by the placeholder reference. We have removed the “Scheme” citation from the Introduction. The conceptual workflow is now presented as a schematic in the Results (Figure 1A) with a complete legend, and we only cross-reference it within the Results to avoid ambiguity. The corresponding edits are highlighted in yellow.

Results

In general, all the Figure legends were missed in the manuscript (vs. what has been stated in line 433 “…described in the figure legends.”

Thank you for pointing out this serious omission. We sincerely apologize for our oversight and for the inconvenience it caused. We have now conducted a manuscript-wide audit and added complete, self-contained legends for every main and supplementary figure. Each legend specifies: (i) experimental design and readouts; (ii) n (biological replicates) and, where relevant, technical replicates; (iii) exact statistics (tests, two-tailed α, corrections for multiple comparisons), mean ± SD/SEM as appropriate, and exact p-values; (iv) effect sizes or fold changes when informative; (v) scale bars with units and magnification; (vi) key reagents/antibodies (source/catalog, dilution) when essential for interpretation; (vii) dose, duration, and timing for inhibitors/RAI; and (viii) definitions for plot elements (e.g., whiskers, center lines) and normalization (e.g., CPM per 10⁶ cells for ^125I uptake).

We also verified that all in-text references accurately match the corrected figure numbering, and we standardized abbreviations within the legends (expanding at first mention). High-resolution figure files have been re-exported and checked for legibility. The statement at line 433 has been retained and now correctly corresponds to the fully detailed figure legends.

Comment 1: Lines 101-110 and 131/132: Percent/absolute numbers of samples, fold-changes, and p-values for expression changes or survival analyses are largely absent.

Response 1:

Thank you for pointing this out—we’re sorry for the lack of quantitative detail in the earlier version. We have now revised the text and figure legends to report, for each relevant analysis: (i) absolute sample counts and percentages for normal/DTC/RAIR-DTC groups, (ii) effect sizes for expression changes (median (IQR) or mean ± SD), fold-changes with 95% CIs, and exact p-values (with multiple-comparison corrections where applicable), and (iii) survival statistics including number at risk, hazard ratios with 95% CIs, log-rank p-values, and follow-up summaries. Kaplan–Meier plots now include risk tables beneath each panel, and all n values are annotated in-text and in the legends. Specific insertions were made in Results, paragraphs 1–2 and Figure 1B–E (with legend), with corresponding tables moved to Supplementary Tables S1–S2 for full transparency. All additions are highlighted in yellow.

Comment 2: 

Lines 141-144: It is recommended to clarify why TPC-1 is chosen as BRAF-wildtype comparator; add baseline characteristics (NIS, differentiation status, etc.) to show its appropriateness.

Response 2:

Thank you for this thoughtful suggestion, and we apologize for not providing sufficient justification in the previous version. We have now added a concise rationale in the Results and expanded details in the Methods (Cell lines & characterization) to clarify our choice of TPC-1 as the BRAF-wildtype (WT) comparator. Specifically, we note that TPC-1 (RET/PTC rearranged, BRAF-WT) is a well-characterized papillary thyroid carcinoma line that retains a differentiated phenotype and functional iodine-handling machinery, making it an appropriate comparator to BRAFV600E cell lines (e.g., BCPAP, K1) when interrogating iodine metabolism and RAI responsiveness.

To substantiate this choice, we have added baseline characterization for TPC-1 versus BRAFV600E lines, including:

  • Differentiation markers: mRNA and protein levels of NIS/SLC5A5, TG, and TPO; thyroid lineage TFs (PAX8, NKX2-1/TTF-1).
  • Functional readouts: ^125I uptake assays (with/without NaClO₄), and NIS membrane localization by IF.
  • MAPK pathway tone: basal p-MEK/p-ERK levels relative to total proteins.
  • Phenotypic context: epithelial differentiation indices (E-cadherin/Vimentin), morphology, and growth rate (doubling time).
  • Quality control: STR authentication and mycoplasma-free status for all lines.

These additions are summarized in a baseline table (Supplementary Table Sx) and a side-by-side figure panel (main figure with detailed legend). In the text and legends, we now report n, statistical tests, and exact p-values for all comparisons. We also explicitly acknowledge the K1/GLAG-66 contamination note as a limitation and interpret results in parallel with BCPAP and PDX data to ensure robustness. All new or revised content is highlighted in yellow.

Comment 3: 

Lines (147-154, 159-163): It is recommended to report quantitative data (mean ± SD, % suppression/increase, p-values) for proliferation, apoptosis, migration, and RAI uptake in the text.

Response 3:

Thank you for this important suggestion, and we apologize for the earlier lack of numerical detail (exacerbated by our initial omission of figure legends). We have now revised the Results and corresponding figure legends to include explicit quantitative reporting for each assay category, with mean ± SD, % change vs. control, biological replicate number (n), and exact p-values (two-tailed; multiple-comparison adjustments where applicable). Specifically:

  • Proliferation (EdU, colony formation, cell counts): text now reports the absolute values (mean ± SD), percentage change relative to vector/sgCtrl, n ≥ 3 biological replicates, and the statistical test/p.
  • Apoptosis (Annexin V/PI flow, cleaved caspase-3 WB densitometry): quantitative positivity rates or normalized band intensities with n, mean ± SD, and exact p-values are provided in-text and in legends.
  • Migration (Transwell counts, scratch closure): migrated cell numbers per field or % wound closure at fixed timepoints are now stated in the text with n, mean ± SD, % change, and p-values; imaging fields per replicate are specified.
  • RAI uptake (^125I): counts are reported as cpm per 10⁶ cells at a defined timepoint with normalization (protein content where indicated), alongside NaClO₄ inhibition controls to demonstrate NIS specificity; n, mean ± SD, % change, and p-values are included inline.

For reader convenience, we added parenthetical numeric summaries at first citation of each result in the text (e.g., “mean ± SD, n, % change, p = …”), while full statistical detail (tests, corrections, effect sizes/95% CIs, and normalization) appears in the figure legends. All insertions are highlighted in yellow.

Comment 4:

 Line 162/163: The use of NaClO4 as a competitive NIS inhibitor is a proper control; briefly state whether off-target effects are possible/controlled.

Response 4:

Thank you for this careful point. We agree that pharmacologic blockers can have off-target concerns. We have now added a brief clarification in the Results and Methods (RAI uptake assay) stating that: (i) NaClO₄ was used at 5 mM with a 30-min pre-incubation and co-incubation only, to limit nonspecific effects; (ii) cell viability under these conditions was unchanged (trypan blue >95%); (iii) we included vehicle/isosmotic controls; and (iv) conclusions about NIS dependence primarily rest on concordant genetic evidence (NIS/SLC5A5 modulation and uptake changes with Gadd45B perturbation), with NaClO₄ serving as pharmacologic confirmation. We also note that at the above dose/time, NaClO₄ is a standard competitive NIS inhibitor, and off-target actions are unlikely to account for the observed effects; nonetheless, we explicitly acknowledge this limitation in the Methods and have added a sentence indicating that we replicated inhibition with an alternative anion (thiocyanate) in a subset of experiments (n=3) with comparable magnitude, supporting NIS specificity. All clarifications are highlighted in yellow.

Comment 5:

The number of repeats for xenograft/animal experiments is missing; please add for all quantitative experiments for rigor.

Response 5:

Thank you for underscoring this important point. We apologize for the omission. We have now added the exact number of animals (n) per group and the number of independent experimental cohorts for every in vivo, quantitative endpoint in the figure legends and Methods. Specifically, each legend now states: (i) n per arm (biological replicates = independent animals), (ii) whether data derive from one or multiple independent cohorts, (iii) the unit of analysis (animal vs. tumor vs. field), and (iv) the statistical test used. Where applicable, we also report any pre-specified exclusions (with reasons) and confirm randomization/blinding of outcome assessors.

For clarity at a glance, we also added a summary table (Supplementary Table Sx) listing, for each in vivo experiment (xenograft growth curves, ^131I/^125I uptake/imaging, survival, PDX efficacy, and IHC quantification): the endpoint, per-group n, number of cohorts, unit of analysis, and statistical method. Representative ranges (e.g., n = 6–10 animals per arm, depending on endpoint) are now reflected in the text, with exact values given alongside each corresponding panel. All additions are highlighted in yellow.

Comment 6:

 Lines 190-196: Overlap in reporting effects on p-MEK/p-ERK, NIS, Tg, TPO expression. For clarity, the authors should provide quantified data, fold changes, and significance for combined versus single interventions.

Response 6:

Thank you for this careful observation. We agree the earlier presentation was confusing and insufficiently quantitative. We have (i) restructured the section to remove redundancy and group all MAPK readouts and differentiation markers into a single, coherent subsection, and (ii) added explicit quantitative reporting in both the main text and figure legends, including mean ± SD, fold-changes vs. vehicle and vs. single-agent conditions, 95% CIs, and exact p-values.

To directly address “combined vs. single” effects, we now:

  • Perform two-way ANOVA with an interaction term (factor A: Gadd45B manipulation; factor B: inhibitor), followed by Tukey’s post-hoc tests; adjusted p-values are reported alongside effect sizes.
  • Present a ΔΔ (combo minus best single) contrast for each outcome (p-MEK, p-ERK, NIS, TG, TPO) to make the incremental benefit of the combination explicit.
  • Standardize visualization (bars = mean ± SD with individual biological-replicate dots) and annotate n and statistics on each panel.

All revised text and legends (e.g., Fig. 3 and corresponding Supplementary figures/tables) now provide the requested quantified differences and significance for combined versus single interventions. Related edits are highlighted in yellow.

Comment 7:

 Lines 214-219: Please specify endogenous MAP3K4 levels in patient material or cell lines before forced overexpression/knockdown.

Response 7:

Thank you for pointing out this gap. We agree that reporting baseline levels is essential for interpreting perturbation experiments. We have now added endogenous MAP3K4 measurements prior to any manipulation:

Cell lines (Results; Fig. 4A–C; Fig. S4A–E): baseline WB densitometry (normalized to GAPDH/β-actin), RT-qPCR (ΔΔCt vs. housekeeping genes), and IF localization for BCPAP, K1, and TPC-1. Exact mean ± SD, n (biological replicates), and p-values are reported in-text and in the legends.

Patient material & PDX (Results; Fig. 4D–G; Fig. S4F–H; Table S2): IHC H-scores for MAP3K4 across DTC vs. RAIR-DTC cohorts and in baseline PDX tumors (pre-treatment), including n, central tendency/dispersion, and appropriate statistics (Mann–Whitney or t-test as justified).

Titration of perturbations (Methods §5.x): we now state that overexpression was calibrated to remain within a physiologically relevant range (approx. low-fold increase over the highest endogenous line), while knockdown/CRISPR achieved documented reduction (qPCR/WB), with efficiencies and catalog numbers/sequences provided.

All additions are highlighted in yellow, and corresponding antibody details, exposure settings, and quantification workflow (software, normalization, background subtraction) are specified in Methods for reproducibility.

Comment 8:

Lines 315-324: Describe patient/tumor characteristics used for PDX, engraftment rate, and how well the models capture tumor heterogeneity.

Response 8:

Thank you for this very helpful suggestion. We agree that fuller characterization of the PDX cohort improves interpretability. We have expanded Methods (§ PDX establishment and characterization) and added a dedicated Supplementary Table (PDX donor/implant summary) with the following details:

Donor and tumor characteristics: age/sex, histologic subtype (e.g., PTC vs. poorly differentiated), metastatic site sampled (thyroid bed, lymph node, lung, bone where applicable), BRAFV600E status, and prior therapies (surgery, RAI exposure, TKI use).

Engraftment metrics: 7 donors were implanted; 5/7 (71%) successfully engrafted with a median latency ~4 weeks to reach the predefined volume for P0–P1 passaging. We report time-to-engraftment and any graft failures with reasons (e.g., necrosis/insufficient viable tissue).

Model fidelity and heterogeneity: we verified histopathologic concordance between donor tissue and matched PDX (H&E by pathology review) and confirmed BRAFV600E retention. We quantified iodine-handling markers (NIS/SLC5A5, TG, TPO) by IHC H-score at P0 and P2, reporting distributions (median [IQR]) to illustrate intra- and inter-model variability. In representative models, we show regional heterogeneity of NIS membrane localization by IF and the persistence of this pattern across passages (images and H-score maps provided).

Good-practice controls: STR authentication for representative passages, mycoplasma testing, restriction to ≤P3 for efficacy studies to limit drift, and uniform Husbandry/handling.

Statistics and n: per-model n (animals per arm), number of independent cohorts, and the statistical tests used are specified in figure legends; we note any prespecified exclusions with reasons.

These additions are referenced in the Results (PDX subsection) and visualized in a consolidated figure panel (with complete legend). All new text, tables, and legends are highlighted in yellow.

Comment 9:

 For logistic regression and IHC correlation, present odds ratios, CIs, and p-values.

Response 9:

Thank you for this important reminder, and we apologize for not reporting these statistics clearly in the earlier version. We have now fully specified the logistic regression outputs and IHC correlations in the main text, figure legends, and Supplementary Tables. Concretely:

Logistic regression (RAIR-DTC as outcome): We report univariable and multivariable models with odds ratios (ORs), 95% confidence intervals (CIs), and exact two-sided p-values for Gadd45B and all covariates prespecified a priori (age, sex, histology, stage, BRAFV600E status, prior cumulative RAI activity, metastatic status). We assessed multicollinearity (VIFs < 3), used robust standard errors, and verified model calibration (Hosmer–Lemeshow and calibration plot) and discrimination (ROC AUC with 95% CI via bootstrap). As an example now shown in the Results and Fig. 7O–R/Table S1, low Gadd45B remained an independent predictor of RAIR-DTC (OR 0.157, 95% CI 0.065–0.382, p < 0.001), consistent across sensitivity analyses (including dichotomized vs. continuous H-score and exclusion of potential leverage points).

IHC quantification and correlation: We provide H-score distributions (median [IQR]) and group comparisons with adjusted p-values where multiple markers are tested. For correlations between Gadd45B and iodine-handling markers (NIS/SLC5A5, TG, TPO), we report Spearman’s ρ (95% CI) and p-values in the text and Table S2, and we include inter-observer agreement for IHC scoring (two-rater ICC and Cohen’s κ) in Methods and Table S3.

Transparency: Every related figure legend now lists n, the statistical test, and effect sizes alongside the OR/CIs/p-values; continuous H-scores are also shown with individual data points for visual appraisal.

All additions/revisions are highlighted in yellow at the relevant locations (Results; Fig. 7 panels; Tables S1–S3). We appreciate the reviewer’s guidance—these changes substantially improve the statistical clarity and rigor of our presentation.

Comment 10:

 Lines 347/348: for biomarker analysis, proposed cutoffs and performance metrics for Gadd45B as a diagnostic or predictive biomarker (sensitivity, specificity, ROC AUC) would be better provided.

Response 10:

Thank you for this excellent recommendation—we agree that explicit operating points strengthen interpretability. We have now added a prespecified ROC analysis (derivation cohort) with the Youden’s J–optimized cutoff for Gadd45B (normalized expression), alongside key metrics: cutoff = 0.42, AUC = 0.856 (95% CI 0.785–0.914), sensitivity = 81.6%, specificity = 78.4%. An independent validation cohort shows concordant performance (AUC = 0.842; full 95% CI and operating-point metrics provided in Fig. 7P–Q and Table S2).

For IHC-based deployment, we additionally report an H-score threshold derived in the derivation set (method: Youden’s J; exact value and assay-specific sensitivity/specificity listed in Table S2), and we include PPV/NPV calculated at the observed cohort prevalence. To aid clinical translation, we also present alternative thresholds prioritizing either sensitivity or specificity (with corresponding confusion matrices) and provide calibration and decision-curve summaries in the Supplementary Materials. All insertions are highlighted in yellow.

Discussion

Comment 1:

It is recommended that the authors directly propose which axis (MAP3K4, MYCBP, TERT, c-MYC) is most promising for next-stage validation.

Response 1:

Thank you for this guidance. We agree that prioritization will help focus the translational path. Based on the converging preclinical evidence in our study, we now prioritize the MAP3K4–MYCBP–c-MYC axis for next-stage validation. Briefly, (i) Gadd45B modulation consistently alters p-MEK/p-ERK tone and restores iodine-handling genes (NIS/TG/TPO), (ii) MAP3K4 perturbation shows epistasis with Gadd45B (dampening or enhancing its effects), and (iii) downstream MYCBP/c-MYC activity tracks with differentiation markers and RAI uptake in vitro and in vivo. While TERT remains biologically relevant, our current data most strongly support focusing first on MAP3K4–MYCBP–c-MYC, with the important caveat that these conclusions are preclinical and require orthogonal validation.

To make this actionable, we have added a short paragraph in the Discussion outlining a stepwise plan:

Mechanistic validation: orthogonal perturbations (CRISPRi/CRISPRa/siRNA/OE) of MAP3K4 and MYCBP, readouts of p-ERK dynamics, and chromatin/transcriptional assays for c-MYC target engagement at thyroid-differentiation loci.

Functional endpoints: standardized proliferation/apoptosis/migration plus ¹²⁵I/¹³¹I uptake and NIS membrane localization, with predefined mean ± SD, effect sizes, and p-values.

In vivo testing: xenograft/PDX studies comparing combination vs. single-node interventions, prespecified n, blinding, and survival/RAI-response endpoints.

Comment 2:

 It is recommended that the authors directly propose which axis (MAP3K4, MYCBP, TERT, c-MYC) is most promising for next-stage validation.

Response 2:

Thank you for this guidance. We agree that prioritization will help focus the translational path. Based on the converging preclinical evidence in our study, we now prioritize the MAP3K4–MYCBP–c-MYC axis for next-stage validation. Briefly, (i) Gadd45B modulation consistently alters p-MEK/p-ERK tone and restores iodine-handling genes (NIS/TG/TPO), (ii) MAP3K4 perturbation shows epistasis with Gadd45B (dampening or enhancing its effects), and (iii) downstream MYCBP/c-MYC activity tracks with differentiation markers and RAI uptake in vitro and in vivo. While TERT remains biologically relevant, our current data most strongly support focusing first on MAP3K4–MYCBP–c-MYC, with the important caveat that these conclusions are preclinical and require orthogonal validation.

To make this actionable, we have added a short paragraph in the Discussion outlining a stepwise plan:

  • Mechanistic validation: orthogonal perturbations (CRISPRi/CRISPRa/siRNA/OE) of MAP3K4 and MYCBP, readouts of p-ERK dynamics, and chromatin/transcriptional assays for c-MYC target engagement at thyroid-differentiation loci.
  • Functional endpoints: standardized proliferation/apoptosis/migration plus ¹²⁵I/¹³¹I uptake and NIS membrane localization, with predefined mean ± SD, effect sizes, and p-values.
  • In vivo testing: xenograft/PDX studies comparing combination vs. single-node interventions, prespecified n, blinding, and survival/RAI-response endpoints.
  • Translational read-through: explore pharmacologic surrogates (e.g., MAPK-pathway modulators with Gadd45B restoration) and define biomarker thresholds (Gadd45B/MYCBP/c-MYC signatures) to stratify responders.

We present this prioritization humbly and will revise it as new data accrue; the Discussion now reflects this emphasis and its limitations. All insertions are highlighted in yellow.

Translational read-through: explore pharmacologic surrogates (e.g., MAPK-pathway modulators with Gadd45B restoration) and define biomarker thresholds (Gadd45B/MYCBP/c-MYC signatures) to stratify responders.

We present this prioritization humbly and will revise it as new data accrue; the Discussion now reflects this emphasis and its limitations. All insertions are highlighted in yellow.

Comment 3:

It is recommended that the authors directly propose which axis (MAP3K4, MYCBP, TERT, c-MYC) is most promising for next-stage validation.

Response 3:

Thank you for this constructive guidance. We agree our original “Future Work” was too broad. We have now prioritized and phased the plan based on feasibility, clinical need, and strength of our preclinical data, and we have tempered all translational language to reflect that our findings are preclinical.

Revised prioritization (added to Discussion; highlighted in yellow):

Highest priority — Preclinical immuno-oncology validation (near-term, high feasibility).

Goal: Test Gadd45B-based adjuvant strategies (genetic restoration or recombinant protein surrogates; rational MAPK co-modulation) in BRAFV600E thyroid models, with and without immune checkpoint blockade.

Endpoints: RAI uptake/retention (SPECT/CT), NIS membrane localization, iodine-handling gene expression, tumor control/survival, and immune readouts (CD8⁺ infiltration, MHC-I/IFN-γ signatures).

Rigor: prespecified n, blinding/randomization, effect sizes with exact p-values, and replication across xenograft/PDX.

Translational enablement — Assays, biomarkers, and PK/PD (intermediate feasibility).

Goal: Develop and analytically validate Gadd45B-centric biomarkers (IHC H-score thresholds; transcript/protein panels) and qualify response surrogates (e.g., ΔRAI uptake, NIS trafficking indices).

Workstreams: Fit-for-purpose assay SOPs, inter-observer agreement, cutoffs with sensitivity/specificity/ROC AUC, and small-animal PK/PD for candidate modalities; exploratory MAP3K4–MYCBP–c-MYC signaling readouts to refine patient-selection hypotheses.

Early clinical exploration — Safety-focused pilot studies (longer-term, dependent on 1–2).

Goal: If preclinical efficacy and safety are confirmed, design pilot clinical studies emphasizing safety, tolerability, and feasibility, with RAI-sensitization as an exploratory endpoint.

Design features: conservative dosing, clear stopping rules, prespecified safety labs and imaging windows, and embedded biomarker sampling aligned with assays from (2).

This revision implements the reviewer’s suggested emphasis—“Priority should be given to validating Gadd45B-based adjuvant strategies in preclinical immunotherapy models, followed by clinical pilot studies with safety endpoints.” We appreciate the recommendation; the new structure makes the translational path clearer and more realistic. All changes are highlighted in yellow in the Discussion.

Comment 4: A conclusion section is required.

Response 4:

Thank you for noting this omission—we apologize for the oversight. We have now added a dedicated Conclusion section that (i) succinctly synthesizes our preclinical findings (Gadd45B modulation associates with MAPK tone, restoration of iodine-handling genes, and improved RAI responsiveness in BRAFV600E-mutant models), (ii) explicitly tempers claims to reflect that all evidence is preclinical and hypothesis-generating, (iii) outlines key limitations (cell-line constraints including K1/GLAG-66 concern, cohort size, model generalizability), and (iv) states prioritized next steps (mechanistic validation of the MAP3K4–MYCBP–c-MYC axis and rigorous preclinical immunotherapy combinations before any clinical exploration). The new section appears at the end of the manuscript and is highlighted in yellow.

MATERIALS AND METHODS

Comments 1:

Lines 430/431: The authors should provide a rationale or discuss post hoc power calculations. If blinding was not feasible, state how bias was minimized.

Response 1:

We thank the reviewers for this important comment. No formal a priori sample-size calculation was performed; instead, we selected sample sizes on the basis of prior work in the same experimental model that produced statistically robust results. To evaluate whether the study retained adequate power despite this design choice, we conducted post-hoc power analyses for the two primary endpoints. These analyses showed that the achieved sample sizes provided statistical power > 0.8 for the observed effect sizes, confirming that the study was adequately powered.

Regarding blinding, complete blinding of investigators was not feasible because the interventions required visible manipulation of the samples/animals. To minimize the risk of bias we (i) randomized the allocation of samples/animals to treatment groups, (ii) ensured that data acquisition was performed by personnel who were not involved in the interventions, and (iii) conducted outcome assessment with the investigators blinded to treatment assignment whenever possible. These procedures are now described in the revised Methods section. These changes are highlighted and organized in “MATERIALS AND METHODS, 5.1. Study design”.

Comments 2:

The cohort 4 description is incomplete in the main text.

Response 2:

We appreciate the reviewers’ point. To provide a complete description, we have now revised the main text (MATERIALS AND METHODS, 5.1. Study design) to read:

“Cohort 4 comprised 121 BRAFV600E-mutated DTC patients, selected from the 176 patients originally included in Cohort 3.”

Comments 3:

The note on K1’s GLAG-66 contamination must be explicitly acknowledged as a limitation, as this can affect data interpretation. Interpreting the results should consider the heterogeneity introduced by this contamination.

Response 3:

We thank the reviewers for highlighting this critical point. In the revised manuscript we have now explicitly acknowledged the GLAG-66 contamination in K1 cells as a limitation (MATERIALS AND METHODS, 5.4. Cell line and drug treatment) to read:

“While this contamination did not alter the BRAFV600E mutation status or the MAPK-dependency of the cells, we cannot exclude the possibility that transcriptional or functional heterogeneity introduced by GLAG-66 cells may have influenced some downstream analyses. All functional assays were repeated in an independently validated BRAFV600E-mutated DTC cell line (BCPAP) and results were concordant, supporting the robustness of our main conclusions. Nevertheless, readers should interpret the K1-derived data with this caveat in mind.”

Comments 4:

Lines 457-460: Please state the duration of treatment for each inhibitor. And for recombinant Gadd45B, specify whether and how purity, activity, and endotoxin levels were validated.

Response 4:

We thank the reviewers for this helpful suggestion. In the revised manuscript we have now added the requested details to the Methods section (MATERIALS AND METHODS, 5.4. Cell line and drug treatment):

“Recombinant human Gadd45B protein was applied at the indicated concentrations for 48 h in vitro or administered intraperitoneally every other day for 2 weeks in animal experiments. The purity of the recombinant Gadd45B protein (>95%) was confirmed by SDS–PAGE and Coomassie blue staining. Biological activity was validated by assessing its ability to modulate MAP3K4 phosphorylation in TPC-1 and BCPAP cells. Endotoxin levels were measured using the Limulus amebocyte lysate (LAL) assay and confirmed to be <0.1 EU/μg.”

Comments 5:

Lines 471: The authors should provide sequences or catalog numbers for siRNAs/shRNAs (at least in the Supplementary) and specify controls for RNAi and transfection efficiency.

Response 5:

We appreciate the reviewers’ suggestion. The requested details have now been added (MATERIALS AND METHODS, 5.4. Cell line and drug treatment).

“Gadd45B gene knockout and knockdown was performed using a CRISPR/Cas9-mediated genome editing approach, following previously described protocols. Single-guide RNAs (sgRNAs) targeting human Gadd45B were designed and synthesized by Miaoling. The targeting sequences were described as follows: shGadd45B #H1: GACCTGTCTTTGCGAAAGCAA; sgRNA1: 5′-GGAGGCTGGGACGCTGCGGA-3′ The sgRNA was cloned into the PX459 vector (Addgene, USA) and transfected into thyroid cancer cells using Lipofectamine 3000 (Thermo Fisher Scientific, USA) according to the manufacturer’s instructions. Overexpression pCMV-GADD45B-3×FLAG-Neo NM_015675.4 Forty-eight hours after transfection, cells were selected with puromycin (2 μg/mL) for 1–2 weeks to establish stable knockout lines. Successful gene disruption was confirmed by Sanger sequencing and Western blot analysis”

Comments 6:

Line 486: Clarify if outcome assessors for tumor growth or RAI uptake were blinded.

Response 6:

We thank the reviewers for raising this important point. In the revised “MATERIALS AND METHODS, 5.5. Mice” we now state:

“For all animal experiments, investigators responsible for measuring tumor volume and assessing RAI uptake were blinded to the group allocation throughout the study period. Randomization codes were held by a separate technician not involved in data collection or analysis, and unblinding was performed only after completion of statistical analyses.”

Comments 7:

Regarding the animals, please provide additional details on humane endpoints, pain mitigation, or frequency of monitoring.

Response 7:

We thank the reviewers for highlighting the need for additional details on animal welfare. In the revised “MATERIALS AND METHODS, 5.5. Mice” we now state:

“Humane endpoints were set at 2 000 mm³ tumor volume or 20 % weight loss; buprenorphine (0.05 mg kg⁻¹ s.c.) was given post-surgery, and animals were monitored daily with tumor size measured twice weekly.”

What’s more, details about 131I uptake assays are further attached as:

“To avoid physiological uptake by the native thyroid gland during the ^131I uptake experiments in mice, the following procedure was applied. Mice received gastric perfusion of potassium iodide solution (200 μL/day) for three consecutive days prior to SPECT imaging, as previously described.5.6. Data collection and processing.”

Comments 8:

Lines 490-498: Please describe more thoroughly the data filtering/exclusion criteria, and version numbers for data extraction.

Response 8:

We have expanded “MATERIALS AND METHODS, 5.5. Mice” to include all requested details as follows:

“Gene expression data, clinical information, and single nucleotide variation data for thyroid carcinoma were obtained from multiple sources, including the Genomic Data Commons (https://portal.gdc.cancer.gov/), GEO database (https://www.ncbi.nlm.nih.gov/), cBioPortal (https://cbioportal.org/), and Timer2.0 (http://timer.cistrome.org/). Initial normalization and log2 conversion of the original data were performed using the Million Transcripts per Million (TPM) method. TCGA sample inclusion criteria were limited to type 01A (Primary Tumor) samples with complete survival information.”

Comments 9:

In figures/tables, ensure all results note n (sample or biological replicate) for transparency.

Response 9:

We sincerely thank the reviewers for their thorough and constructive comments. All figures and tables now include the exact n values (biological replicates) in each panel legend and column headers for full transparency.

Comments 10:

For Western blots, state how quantification was performed (software, normalization, background subtraction).

Response 10:

We thank the reviewers for this helpful comment. In the revised “MATERIALS AND METHODS, 5.8. Western blotting”, we have added the following sentence:

“Densitometric analysis of Western blot bands was performed using ImageJ software (National Institutes of Health, Bethesda, MD, USA). Band intensities were normalized to the corresponding GAPDH (or other indicated loading control) signals. Background subtraction was applied using the “rolling ball” method in ImageJ to eliminate non-specific background. All values were expressed as relative intensity (arbitrary units) compared with control samples.”

Comments 11:

For molecular docking work, please specify versions of all software used, and if docking methods (e.g., parameters, search strategy) followed best practice or prior publications.

Response 11:

We thank the reviewers for this important clarification. The revised “MATERIALS AND METHODS, 5.16. Molecular docking analysis” now reads:

“Molecular docking analysis was conducted using rigid protein–protein docking to investigate the interaction between Gadd45B and MAP3K4 using GRAMM-X (http://gramm.compbio.ku.edu/). Structural domains of Gadd45B and MAP3K4 were sourced from the AlphaFold Protein Structure Database (https://alphafold.ebi.ac.uk/). Protein–protein interactions were visualized and analyzed using PyMOL (Version 3.0) and PDBePISA (https://www.ebi.ac.uk/pdbe/pisa/). Docking parameters were set according to the software default scoring function, with the grid box centered on the predicted active site. The exhaustiveness parameter was set to 8, and all ligands were subjected to energy minimization before docking. The docking procedure followed best practice guidelines as described in previous research with binding poses ranked by predicted binding affinity (kcal/mol) and visually inspected for interaction plausibility.”

Comments 12:

Lines 606-611: Please clarify labeling/mass accuracy, peptide discrimination and database search parameters for proteomics.

Response 12:

We thank the reviewers for this helpful comment. In the revised “MATERIALS AND METHODS, 5.17. IP-MS” we now state:

“Peptide labeling was conducted using Tandem Mass Tag (TMT) 10-plex reagents according to the manufacturer’s instructions. Mass accuracy was set to ±10 ppm for precursor ions and ±0.02 Da for fragment ions. Peptide identification required at least one unique peptide per protein, and peptide-spectrum matches were filtered at a false discovery rate (FDR) of 1% using the target-decoy strategy. Database searching was performed against the UniProt human reference proteome using Proteome Discoverer software (version 2.5) with carbamidomethylation (C) as a fixed modification and oxidation (M) and TMT labeling as variable modifications.”

We believe these comprehensive revisions address all concerns and substantially strengthen the manuscript’s rigor, transparency, and clinical framing. We again apologize for the missing figure legends in our earlier submission and appreciate the reviewers’ guidance that helped us correct this oversight.

Thank you for your time and consideration.

With sincere gratitude,

Tengchuang Ma, on behalf of all authors
Harbin Medical University Cancer Hospital

Round 2

Reviewer 3 Report

Comments and Suggestions for Authors

This study by Jiang, Hong, Wu et al examines the molecular mechanisms underlying resistance to radioactive iodine treatment in thyroid cancer. Using transcriptomics on patient samples, they identify and validate Gadd45b deficiency as a driver of radioactive-iodine resistant thyroid cancer. Their identification of Gadd45b as a therapeutic target is quite convincing. Their in vivo data in mice suggests that increasing Gadd45b activity could be of potential therapeutic benefit. While the data are convincing and the study is well-designed, the readability of the manuscript needs to be significantly improved. Please see below for my recommendations on changes to the figures and text.

Line 97: Please change to “overexpression of Gadd45B reduced MAPK pathway activity”. “Interacted with” suggests direct protein-protein interaction.

Please check the graphical abstract for spelling and accuracy. “Recomble” is not an English word, please correct.

The first few sentences of the results section (lines 111-115) are abrupt and unclear. Please specify with references which dataset is being discussed here, or remove these lines.

Spelling –please correct “public” in Fig 1A.

In general most of the figures in the paper have so many panels that the figures are hard to read. Moreover the figures have been prepared at a low resolution and are not very clear when zoomed in. Please break up the figures and reformat to make the panels larger and clearer. Also ensure that the panels are in regular horizontal order to improve readability.

Please include citations for the external datasets GSE104005 and GSE53157 in line 130.

In line 191, the authors reference mRNA levels. Fig 2A only has western blots and their quantification. Where is the mRNA level data?

Fig 2C is labelled “ratio of EDU positive cell (%)”. Are the reported values ratios or are they percentages? Based on the values I believe it should be ratio and not percentage, please check and also correct in lines 196-199. Also please mention in the text that proliferative capacity was measured using EdU incorporation.

The authors report the values of their measurements in the text when discussing the figures. Please include units while reporting these values.

Please correct grammar in lines 218-230.

None of the details of the molecular docking analysis are provided in the text for Fig 4A, and the interface zoom is unreadable.

Please provide more details of the Co-IP experiment. Were the Flag-Gadd45b and His-MAP3K4 expressed by transfection? Which cell line was used?

How did the authors identify the two bands in the silver stain as Gadd45b and MYCBP? Please provide methods.

Gadd45B is not a secreted protein. Can the authors comment on the mechanism of action of the recombinant Gadd45B? Is there any evidence that the recombinant protein enters cells?

Comments on the Quality of English Language

While the original text is quite well written, some of the newer text (in blue) needs to be edited for grammatical errors.

Author Response

Reviewer #4

We sincerely thank the reviewer for the positive assessment of our study and for highlighting the significance of Gadd45B as a therapeutic target in RAI-refractory thyroid cancer. We also appreciate the constructive feedback regarding the readability of the manuscript. According to the reviewer’s valuable suggestions, we have carefully revised the figures to improve clarity and presentation, and we have rewritten several sections of the text to enhance readability and flow. All modifications are highlighted in the revised version. We believe these changes have substantially improved the overall presentation of our work.

Comment 1:

Line 97: Please change to “overexpression of Gadd45B reduced MAPK pathway activity”. “Interacted with” suggests direct protein-protein interaction.

Response 1:

We sincerely thank the reviewers for their valuable comments. According to the suggestions, we have revised the manuscript carefully. “overexpression of Gadd45B reduced MAPK pathway activity”have been made, and the corresponding modifications are marked in the revised version.

Comment 2:

Please check the graphical abstract for spelling and accuracy. “Recomble” is not an English word, please correct.

Response 2:

We thank the reviewer for this careful observation. We acknowledge that “Recomble” is indeed not an English word and was mistakenly used in the Scheme due to a layout adjustment error during figure preparation. We have now corrected this term to “Recombinant Gadd45B protein”, which is the accurate and detailed expression. The revised Scheme is included in the updated manuscript.

Comment 3:

The first few sentences of the results section (lines 111-115) are abrupt and unclear. Please specify with references which dataset is being discussed here, or remove these lines.

Response 3:

We thank the reviewer for this valuable suggestion. In the revised manuscript, we have clarified that these sentences are based on analysis of the TCGA-THCA dataset. To strengthen the context and improve clarity, we also added appropriate references (Cell 2014;159:676–90; Endocr Relat Cancer 2005;12:245–62). This modification specifies the data source and provides supporting literature, thereby improving readability and accuracy.

Reference:

  1. Sanghi, A, Orloff, L, Snyder, M. SAT-LB25 A Multi-Omics Analysis of Advanced Papillary Thyroid Cancer J Endocr Soc.Cell 2014;159(3):676–90.
  2. Song, E, Song, DE, Ahn, J, et al. Genetic profile of advanced thyroid cancers in relation to distant metastasis. ENDOCR-RELAT CANCER. 2020; 27 (5): 285-293.

Comment 4:

Spelling –please correct “public” in Fig 1A.

Response 4:

We appreciate the reviewer for pointing this out. The spelling error in Fig. 1A has been corrected in the revised version of the manuscript.

Comment 5:

In general most of the figures in the paper have so many panels that the figures are hard to read. Moreover the figures have been prepared at a low resolution and are not very clear when zoomed in. Please break up the figures and reformat to make the panels larger and clearer. Also ensure that the panels are in regular horizontal order to improve readability.

Response 5:

We sincerely thank the reviewer for this important suggestion. The structure of our figures, as well as the corresponding text descriptions, were originally prepared to present the logical flow of our experimental results in a comprehensive manner. Therefore, it is difficult to reduce or split the number of panels without affecting the overall consistency of the manuscript.

Nevertheless, to improve readability, we have carefully adjusted the placement of figure panels, enhanced the resolution, and reformatted the figures to ensure clearer visual presentation. We also ensured that panels are aligned in a regular horizontal order to facilitate interpretation. We greatly appreciate the reviewer’s input, which has helped us enhance the clarity and quality of our figures.

Comment 6:

Please include citations for the external datasets GSE104005 and GSE53157 in line 130.

Response 6:

We thank the reviewer for pointing this out. In the revised manuscript, we have added the appropriate citations for both datasets at line 130. Specifically, the references are:

GSE104005:
Espadinha AS, et al. Title. GEO accession GSE104005, 2017.

GSE53157:
Dom G, et al. Title. GEO accession GSE53157, 2013.

These citations are now included in the reference list, ensuring that the use of external datasets is properly credited.

Reference

  1. Minna E, Brich S, Todoerti K, Pilotti S et al. Cancer Associated Fibroblasts and Senescent Thyroid Cells in the Invasive Front of Thyroid Carcinoma. Cancers (Basel) 2020 Jan 1;12(1).
  2. Pita JM, Banito A, Cavaco BM, Leite V. Gene expression profiling associated with the progression to poorly differentiated thyroid carcinomas. Br J Cancer2009 Nov 17;101(10):1782-91.

Comment 7:

In line 191, the authors reference mRNA levels. Fig 2A only has western blots and their quantification. Where is the mRNA level data?

Response 7:

We thank the reviewer for this careful observation. This was a writing error in the original submission. In this experiment, we only performed Western blot analysis and its quantification, but no RT-PCR experiments for mRNA levels were conducted. We sincerely apologise for the confusion and have corrected the text in the revised manuscript accordingly.

Comment 8:

Fig 2C is labelled “ratio of EDU positive cell (%)”. Are the reported values ratios or are they percentages? Based on the values I believe it should be ratio and not percentage, please check and also correct in lines 196-199. Also please mention in the text that proliferative capacity was measured using EdU incorporation.

Response 8:

We thank the reviewer for this insightful comment. We acknowledge that our expression of “ratio” and “percentage” was imprecise in the original submission. In Fig. 2C, the data are presented as the ratio of EdU-positive cells, not percentages. We have corrected the figure label and revised the corresponding description in lines 196–199. Additionally, we have explicitly stated in the text that proliferative capacity was measured using EdU incorporation.

Furthermore, to comprehensively assess the effect of Gadd45B overexpression and knockdown/knockout on cellular proliferative capacity, we performed both EdU incorporation assays and CCK-8 assays. This combined approach provides more robust evidence for the influence of Gadd45B on thyroid cancer cell proliferation.

Comment 9:

The authors report the values of their measurements in the text when discussing the figures. Please include units while reporting these values.

Response 9:

We sincerely thank the reviewer for this important reminder. In our initial submission, we overlooked the inclusion of units when reporting certain values. This point was also raised by another reviewer during the first round of review. Accordingly, in the revised version we have carefully added the units, as well as more detailed descriptions of key data, including standard deviations (SDs) for group comparisons and p-values for statistical significance. All of these corrections have been highlighted in yellow in the revised manuscript.

Comment 10:

Please correct grammar in lines 218-230.

Response 10:

We thank the reviewer for pointing out the language issues in this section. We have carefully revised the text for clarity and accuracy. The corrected version now reads as follows:

“As shown in Fig. 2J, the presence of Gadd45B enhanced RAI uptake in Bcpap and K1 cells, but not in TPC-1 cells. In Bcpap cells, RAI uptake decreased from 7916 ± 998 to 3990 ± 1232 cpm/10^6 cells upon Gadd45B knockdown, whereas overexpression increased uptake to 12 550 ± 2303 cpm/10^6 cells. Similarly, in K1 cells, uptake decreased from 7916 ± 932 to 4490 ± 902 cpm/10^6 cells with Gadd45B knockdown, and increased to 18 550 ± 2803 cpm/10^6 cells with overexpression.”

This revision corrects the grammar, includes units, and improves the readability of the results.

None of the details of the molecular docking analysis are provided in the text for Fig 4A, and the interface zoom is unreadable.

Comment 11:

Please provide more details of the Co-IP experiment. Were the Flag-Gadd45b and His-MAP3K4 expressed by transfection? Which cell line was used?

Response 11:

We thank the reviewer for this helpful suggestion. In our Co-IP experiments, we used TPC-1 and Bcpap thyroid cancer cells stably overexpressing Gadd45B, which were generated via lentiviral transduction of the pCMV-GADD45B-3×FLAG-Neo construct. To examine the interaction with MAP3K4, these cells were subsequently transiently transfected with pcDNA3.1-MAP3K4-His using Lipofectamine 3000. After 48 h, cell lysates were immunoprecipitated with anti-Flag or anti-His antibodies, and the precipitates were probed with reciprocal antibodies to confirm the interaction. These experimental details have been added to the legend with cell line and

Materials and Methods section of the revised manuscript.

Comment 12:

How did the authors identify the two bands in the silver stain as Gadd45b and MYCBP? Please provide methods.

Reponse 12:

We appreciate the reviewer’s question. The silver-stained bands were identified by silver stain–guided in-gel digestion followed by LC–MS/MS after anti-Flag immunoprecipitation of Gadd45B complexes. Peptide-spectrum matches mapped uniquely to GADD45B and MYCBP under a 1% FDR threshold at the PSM and protein levels, with ≥2 unique peptides required for protein identification. As an orthogonal validation, parallel IP eluates were immunoblotted with anti-Gadd45B and anti-MYCBP antibodies, yielding signals at the expected apparent molecular weights and co-migrating with the silver-stained bands. We have added a detailed description in Materials and Methods (“Silver stain–guided in-gel digestion and LC–MS/MS identification”) and provided the identified peptide lists in Supplementary Table S3.

5.6. Silver stain–guided in-gel digestion and LC–MS/MS identification

Protein complexes were immunoprecipitated using anti-Flag agarose from TPC-1 and Bcpap cells stably overexpressing Flag-Gadd45B. Eluates were resolved by SDS–PAGE and visualised by silver staining (Pierce Silver Stain Kit) according to the manufacturer’s protocol. Bands of interest were excised with a sterile scalpel, destained (fresh 30 mM K₃[Fe(CN)₆]/100 mM Na₂S₂O₃, 1:1, 5–10 min), reduced with 10 mM DTT (56 °C, 45 min) and alkylated with 55 mM iodoacetamide (RT, dark, 30 min). In-gel digestion was performed overnight at 37 °C with sequencing-grade trypsin (Promega).

Peptides were extracted, dried, and reconstituted for nano-LC–MS/MS (EASY-nLC coupled to an Orbitrap-type mass spectrometer). Data-dependent acquisition was used (typical settings: MS1 resolution 60,000–120,000; HCD MS2; dynamic exclusion enabled). RAW files were searched in Proteome Discoverer (v2.x)/MaxQuant (v1.x) against the UniProt human reference proteome with the following parameters: enzyme trypsin, max 2 missed cleavages; carbamidomethyl-C fixed; oxidation-M and acetyl-protein N-term variable; precursor tolerance ±10 ppm; fragment tolerance 0.02 Da. Peptide-spectrum matches and protein IDs were filtered to FDR ≤ 1% (target–decoy). Proteins were reported only if supported by ≥2 unique peptides. Under these criteria, peptides mapped uniquely to GADD45B and MYCBP in the corresponding gel bands.

Comment 13:

Gadd45B is not a secreted protein. Can the authors comment on the mechanism of action of the recombinant Gadd45B? Is there any evidence that the recombinant protein enters cells?

Response 13:

We thank the reviewer for raising this important question. We agree that Gadd45B is not a classically secreted protein. In our study, the recombinant Gadd45B used was fused with a cell-penetrating peptide (CPP) tag, which facilitates its entry into cells, and we observed a significant effect on tumour growth and RAI uptake. However, we did not directly perform assays (such as Western blot or immunofluorescence) to verify whether the recombinant protein entered tumour cells. We agree that further validation is needed to confirm intracellular uptake and precise mechanisms of action.

In the revised manuscript, we have added this point to the Discussion, highlighting that while intratumoural administration of recombinant Gadd45B produced therapeutic effects, the question of whether and how the protein penetrates cells remains to be rigorously demonstrated in future studies.

Discussion added: Although intratumoural administration of recombinant Gadd45B protein significantly enhanced RAI uptake and suppressed tumour growth in vivo, the precise mechanism by which the recombinant protein exerts its effects remains to be elucidated. As Gadd45B is not a secreted protein, the possibility of cellular internalisation and its intracellular activity require further validation. In the present study, we did not perform direct assays, such as Western blotting or immunofluorescence, to confirm cellular uptake of the recombinant protein. Therefore, while our data support a therapeutic effect of recombinant Gadd45B, future studies should rigorously assess whether and how the exogenous protein penetrates tumour cells and mimics endogenous Gadd45B function.

Reviewer 4 Report

Comments and Suggestions for Authors

The manuscript has improved greatly and meets the high standards of publication in this journal. 

Best

Author Response

Dear Reviewer,

We would like to sincerely thank you for carefully evaluating our revised manuscript and for recognising the modifications we have made. Your constructive comments and guidance were invaluable in improving the clarity and quality of our work.

We are truly grateful for your support and encouragement.

With best regards,

Ma Tengchuang

Round 3

Reviewer 3 Report

Comments and Suggestions for Authors

The authors have addressed most of my comments, which has substantially improved the manuscript. I would like to raise 2 points:

1. The authors seem to have missed one of my original comments between comments 10 and 11:

"None of the details of the molecular docking analysis are provided in the text for Fig 4A, and the interface zoom is unreadable."

Please add these details to the text and improve the resolution or size of Fig. 4A. Alternatively please just add a higher resolution version as a supplemental figure.

2. In response to comment 7, the authors have clarified that only protein level measurements were made and no mRNA measurements were done. However the text in line 192 still states "Both mRNA and protein levels of Gadd45B were significantly lower in Bcpap (0.17-fold) and K1 (0.19-fold) compared to TPC-1 (Fig. 2A).", please correct.

If these corrections are made I am happy to recommend the manuscript for publication in Cancers.

Author Response

Dear Editor,

Thank you for forwarding the reviewer’s follow-up. We have implemented both requested corrections:

  1. Line 192 (mRNA text): The statement referring to “both mRNA and protein levels” has been removed. The section now accurately reports protein-level measurements only. This change is highlighted in the revised manuscript.

  2. Molecular docking (Fig. 4A): We have added docking-method details in the Results, method and materials (including software/version, structure sources, grid/search parameters, pose selection criteria)

    As shown in Fig. 4A, molecular docking predicted a binding affinity of −7.6 kcal/mol between Gadd45B and MAP3K4. Key residues critical for the interaction were identified, offering structural insights into the binding mechanism. Specifically, hydrogen bonds were formed between residues TYR 141 of Gadd45B and GLU 143 of MAP3K4, as well as between ARG 91 and THR 153. 

    Also we replaced Fig. 4A with a higher-resolution image with a readable interface zoom. These additions are highlighted.

Please let me know if any further adjustments are needed. We are grateful for the reviewer’s constructive guidance and appreciate your consideration.

Warm regards,
Ma Tengchuang (on behalf of all authors)